# Single-cell transcriptome analysis identifies a unique tumor cell type producing multiple hormones in ectopic ACTH and CRH secreting pheochromocytoma

Xuebin Zhang[1†], Penghu Lian[1†], Mingming Su[2†], Zhigang Ji[1], Jianhua Deng[1], Guoyang Zheng[1], Wenda Wang[1], Xinyu Ren[3], Taijiao Jiang[2,4], Peng Zhang[5]*, Hanzhong Li[1]*

[1]Department of Urology, Peking Union Medical College Hospital, Chinese Academy of Medical Sciences & Peking Union Medical College, Beijing, China; [2]Institute of Basic Medical Sciences Chinese Academy of Medical Sciences, School of Basic Medicine Peking Union Medical College, Beijing, China; [3]Department of Pathology, Peking Union Medical College Hospital, Chinese Academy of Medical Sciences & Peking Union Medical College, Beijing, China; [4]Suzhou Institute of Systems Medicine, Jiangsu, China; [5]Beijing Key Laboratory for Genetics of Birth Defects, Beijing Pediatric Research Institute, Beijing Children's Hospital, Capital Medical University, National Center for Children's Health, Beijing, China

*For correspondence:
zhangpengdyx@163.com (PZ);
lihzh@pumch.cn (HL)

†These authors contributed
equally to this work

Competing interest: The authors
declare that no competing
interests exist.

Reviewing Editor: Murim
Choi, Seoul National University,
Republic of Korea

**Abstract** Ectopic Cushing's syndrome due to ectopic ACTH&CRH-secreting by pheochromocytoma is extremely rare and can be fatal if not properly diagnosed. It remains unclear whether a unique cell type is responsible for multiple hormones secreting. In this work, we performed single-cell RNA sequencing to three different anatomic tumor tissues and one peritumoral tissue based on a rare case with ectopic ACTH&CRH-secreting pheochromocytoma. And in addition to that, three adrenal tumor specimens from common pheochromocytoma and adrenocortical adenomas were also involved in the comparison of tumor cellular heterogeneity. A total of 16 cell types in the tumor microenvironment were identified by unbiased cell clustering of single-cell transcriptomic profiles from all specimens. Notably, we identified a novel multi-functionally chromaffin-like cell type with high expression of both POMC (the precursor of ACTH) and CRH, called ACTH+&CRH + pheochromocyte. We hypothesized that the molecular mechanism of the rare case harbor Cushing's syndrome is due to the identified novel tumor cell type, that is, the secretion of ACTH had a direct effect on the adrenal gland to produce cortisol, while the secretion of CRH can indirectly stimulate the secretion of ACTH from the anterior pituitary. Besides, a new potential marker (GAL) co-expressed with ACTH and CRH might be involved in the regulation of ACTH secretion. The immunohistochemistry results confirmed its multi-functionally chromaffin-like properties with positive staining for CRH, POMC, ACTH, GAL, TH, and CgA. Our findings also proved to some extent the heterogeneity of endothelial and immune microenvironment in different adrenal tumor subtypes.

## Editor's evaluation

The study described an extremely rare type of adrenal pheochromocytoma that secretes both ACTH and CRH, in addition to catecholamines. Single-cell RNA sequencing of the tumor and other tumors

revealed a group of cells that are responsible for the hormone secretion. We believe that this work will provide an interesting example of functional endocrine tumors and how they are formed.

## Introduction

Cushing's syndrome (CS) is a rare disorder caused by long-term exposure to excessive glucocorticoids, with an annual incidence of about 0.2–5.0 per million (*Lacroix et al., 2015*; *Newell-Price et al., 2006*; *Lindholm et al., 2001*; *Steffensen et al., 2010*; *Bolland et al., 2011*; *Valassi et al., 2011*). About 80% of CS cases are due to ACTH secretion by a pituitary adenoma, about 20% are due to ACTH secretion by nonpituitary tumors (ectopic ACTH syndrome [EAS]), and 1% are caused by corticotropin-releasing hormone (CRH)-secreting tumors (*Alexandraki and Grossman, 2010*; *Ejaz et al., 2011*; *Ballav et al., 2012*). Most EAS tumors (~60%) are more common intrathoracic tumors, only 2.5–5% of all EAS are caused by a pheochromocytoma (*Alexandraki and Grossman, 2010*; *Isidori et al., 2006*; *Ilias et al., 2005*; *Aniszewski et al., 2001*). Pheochromocytoma, a catecholamine-producing tumor, becomes even rarer when it is capable of both secreting ACTH and CRH (*Lenders et al., 2005*; *Zelinka et al., 2007*). By 2020, only two cases with pheochromocytoma secreted both ACTH and CRH were reported (*Elliott et al., 2021*; *O'Brien et al., 1992*; *Jessop et al., 1987*). As one of the largest adrenal tumor treatment centers in China, our hospital, Peking Union Medical College Hospital (PUMCH) receives more than 500 adrenal surgery performed per year, with almost 100 cases undergoing pheochromocytoma surgery. But so far, we have encountered only one case of pheochromocytoma secreting both ACTH and CRH, which was first reported in this study.

Since the combination of dual ACTH/CRH secreting pheochromocytoma with CS is extremely rare, there is limited knowledge about the diagnosis and management of this disease. Ectopic secretion hormones ACTH and CRH may complicate the presentation of pheochromocytoma, and this tumor usually leads to CS, which can be fatal if not properly diagnosed and managed (*Ballav et al., 2012*; *Ilias et al., 2005*; *Lenders et al., 2014*; *Lase et al., 2020*). Surgical resection of the pheochromocytoma is the primary treatment option. Although previous studies have reported ectopic ACTH and CRH secreting pheochromocytomas, it was unclear whether a unique cell type that produces multiple hormones influences CS. The concept of 'one cell, one hormone, and one neuron one transmitter,' which is known as Dale's Principle (Dale in 1934; for detailed discussion, see *Burnstock, 1976*), has dominated the understanding of neurotransmission for many years (*Burnstock, 1976*). Currently, single-cell RNA-sequencing (scRNA-seq) can examine the expression profiles of a single cell and is recognized as the gold standard for defining cell states and phenotypes (*Tang et al., 2009*; *Tammela and Sage, 2020*; *Kolodziejczyk et al., 2015*; *Patel et al., 2014*; *Tirosh et al., 2016b*; *Tirosh et al., 2016a*; *Puram et al., 2017*; *Venteicher et al., 2017*; *Young et al., 2018*; *Bernard et al., 2019*; *Segerstolpe et al., 2016*; *Reichert and Rustgi, 2011*). It can reveal the presence of rare and novel unique cell types, such as CFTR-expressing pulmonary ionocytes on lung airway epithelia (*Montoro et al., 2018*; *Plasschaert et al., 2018*). It also provides an unbiased method to better understand the diversity of immune cells in the complex tumor microenvironment (*Papalexi and Satija, 2018*; *Stubbington et al., 2017*).

In this study, we reported a rare case of CRH/ACTH-secreting pheochromocytoma infiltrating the kidney and psoas muscle tissue. scRNA-seq identified a unique chromaffin-like cell type, called ACTH+&CRH + pheochromocyte, with both high expression of POMC (precursor for ACTH) and CRH pheochromocyte as well as TH (tyrosine hydroxylase, a key enzyme for catecholamine synthesization). Immunocytochemical and immunofluorescence staining showed all for these markers, which confirmed the tumor capable of multiple hormones secreting characteristics. We determined that the expression of POMC directly causes the secretion of ACTH, and the expression of CRH indirectly promotes the secretion of ACTH hormone, which ultimately leads to CS. After the tumor resection, clinical manifestations also showed complete remission of CS. For comparison, other adrenal tumor subtypes were also collected and studied, namely, a common pheochromocytoma (without ectopic ACTH or CRH secretion function) and two adrenocortical adenomas. We used a scRNA-seq approach to obtain transcriptomic profiles for all collected samples and identified a list of differentially expressed genes (DEGs) through cell clustering and markers finding. Notably, GAL, co-expressed with ACTH and CRH, could be a new candidate marker to detect the rare ectopic ACTH+&CRH + secreting pheochromocytes by comparing ACTH+&CRH + pheochromocyte with common pheochromocyte and cortical

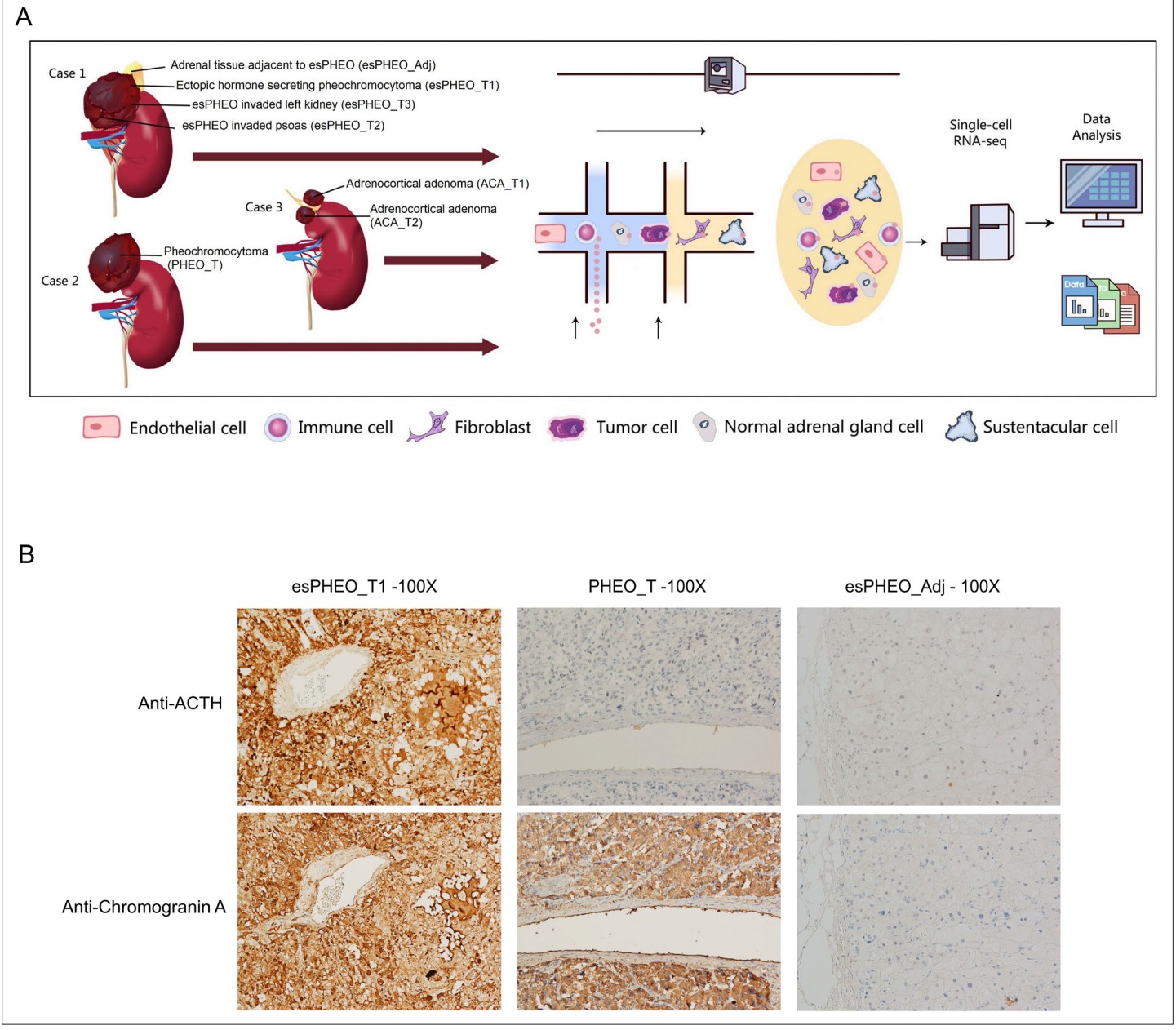

**Figure 1.** Clinical sample collection of adrenal tumor and adjacent specimen for scRNA-seq analysis. (**A**) scRNA-seq workflow for three tumor specimens (esPHEO_T1, esPHEO_T2, and esPHEO_T3) and one adjacent specimen (esPHEO_Adj) from the rare pheochromocytoma with ectopic ACTH and CRH secretion (Case 1), 1 tumor specimen (PHEO_T) from common pheochromocytoma (Case 2), and two tumor specimens (ACA_T1, ACA_T2) from adrenocortical adenoma (Case 3). (**B**) Immunohistochemistry staining results of pheochromocytoma (esPHEO_T1 and PHEO_T) and adjacent specimen (esPHEO_Adj) with chromogranin A (CgA) and ACTH markers.

cell clusters. It suggested that GAL, which encodes small neuroendocrine peptides, may be locally involved in the regulation of the hypothalamic-pituitary-adrenal (HPA) axis.

## Results
### Single-cell profiling and unbiased clustering of collecting specimens

We applied scRNA-seq methods to perform large-scale transcriptome profiling of seven prospectively collected samples from tumors and peritumoral tissue of three adrenal tumor patients (*Figure 1A*). Case 1 suffered from a rare pheochromocytoma with typical Cushingoid features. The laboratory

results showed high levels of cortisol, ACTH, and catecholamines. The abdominal contrast-enhanced computer tomography scanning revealed bilateral adrenocortical hyperplasia and irregular tumor within the left adrenal. After the resection, we collected three dissected tumor specimens (esPHEO_T1, esPHEO_T2, and esPHEO_T3) from different anatomic sites of the tumor and an adrenal tissue adjacent to the tumor (esPHEO_Adj). For comparison, we also collected other adrenal tumors, namely, a common pheochromocytoma (PHEO_T) from Case 2 and two adrenocortical adenomas (ACA_T1 and ACA_T2) from Case 3. Case 2 showed elevated catecholamines and normal levels of cortisol and ACTH. Case 3 showed a high level of cortisol, a low level of ACTH, and an interme-diate level of catecholamines. The detailed clinical information for the three cases was summarized in *Appendix 1—table 1*. To investigate the difference of the secretory function, we performed the immu-nohistochemistry (IHC) staining of selected markers, CgA (chromogranin A) and ACTH in esPHEO_T1, PHEO_T, and esPHEO_Adj samples (*Figure 1B*). We observed that CgA positive cells were present in both pheochromocytomas (esPHEO_T1 and PHEO_T), but ACTH positive cells were only observed in the rare pheochromocytoma (esPHEO_T1) with the ACTH-secreting cellular characteristics. As expected, there were no CgA and ACTH positive cells in the adjacent sample (esPHEO_Adj). Thus, at the clinical stage, our histopathology results confirmed that Case 1 was a rare ectopic ACTH secreting pheochromocytoma which stained positively for both ACTH and CgA.

Then, we applied scRNA-seq approaches to selected seven specimen samples (six tumors and one sample adjacent to the tumor). The tissues after resection were rapidly digested into a single-cell suspension, and the 3′-scRNA-seq protocol (Chromium Single Cell 3′ v2 Libraries) was performed for each sample unbiasedly. After quality control filtering to remove cells with low gene detection, high mitochondrial gene coverage, and doublets filtration, we compiled a unified cells-by-genes expres-sion matrix of a total of 44,511 individual cells (*Supplementary file 1*, *Appendix 1—figure 2*). Then the SCT-transformed normalization, principal component analysis (PCA), was employed to perform unsupervised dimensionality reduction. Then, the cells were clustered based on the graph-based clus-tering analysis, and visualized in the distinguished diagram using the Uniform Manifold Approximation and Projection (UMAP) method. The marker genes were calculated to identify each cell cluster by performing differential gene expression analysis (*Supplementary file 2*).

As shown in *Figure 2A*, the distinct cell clusters were identified and the conventional cell lineage gene markers were employed to annotate the clusters, such as CHGA and CHGB for adrenal chro-maffin cell, cytochrome P450 superfamily for adrenocortical cell, S100B for sustentacular cell, GNLY for NK cell, MS4A1 for B cell, CD8A for CD8+ T cell, and IL7R for CD4+ T cell. Based on the expres-sion of gene markers, we recognized a total of 16 main cell groups: ACTH+&CRH + pheochromo-cyte, pheochromocyte, adrenocortical, sustentacular, erythroblast/granulosa, endothelial, fibroblast, neutrophil, monocyte, macrophage, plasma, B, NK, CD8+ T&NKT, CD8+ T, and CD4+ T, among which the endothelial cell group was composed of four endothelial cell subgroups. The heatmap showed the expression levels of specific cluster markers for each cell phenotype that we identified (*Figure 2B*). For this analysis, we specifically focused on the four types of adrenal cells and showed their markers in a heatmap (*Appendix 1—figure 3*). Additionally, we detected the transcription factors alongside their candidate target genes, which are jointly called regulons. The analysis scored the activity of regulon for each cell (*Appendix 1—figure 4A*) and yielded specific regulons for each cellular cluster (*Appendix 1—figure 4B*). We also specifically focused on the adrenal cells and found XBP1 as the top regulons for ACTH+&CRH + pheochromocyte and adrenocortical cell type (*Appendix 1—figure 4C*).

## Identification of a previously unrecognized cell type

The presence of heterogeneous cell populations in different adrenal tumor specimens and the peritu-moral sample (*Figure 3A*) prompted us to investigate their cellular compositions and characteristics. As shown in *Figure 3B*, different sources of specimens represented distinct cell type compositions. Notably, although the size of the cell clusters of the adrenal gland was relatively small, four distinct subtypes of adrenal cells were observed, including ACTH+&CRH + pheochromocyte, pheochro-mocyte, adrenocortical cells, and sustentacular cells. The ACTH+&CRH + pheochromocytoma cell subtype was specific to three tumor samples, esPHEO_T1, esPHEO_T2, and esPHEO_T3 from Case 1, but was not observed in the peritumoral sample (esPHEO_Adj) and other adrenal tumor samples from Case 2 (PHEO_T) and Case 3 (ACA_T1 and ACA_T2). This result was consistent with the clinical symptoms in our earlier reports that ACTH was only over-secreted in pheochromocytoma of Case

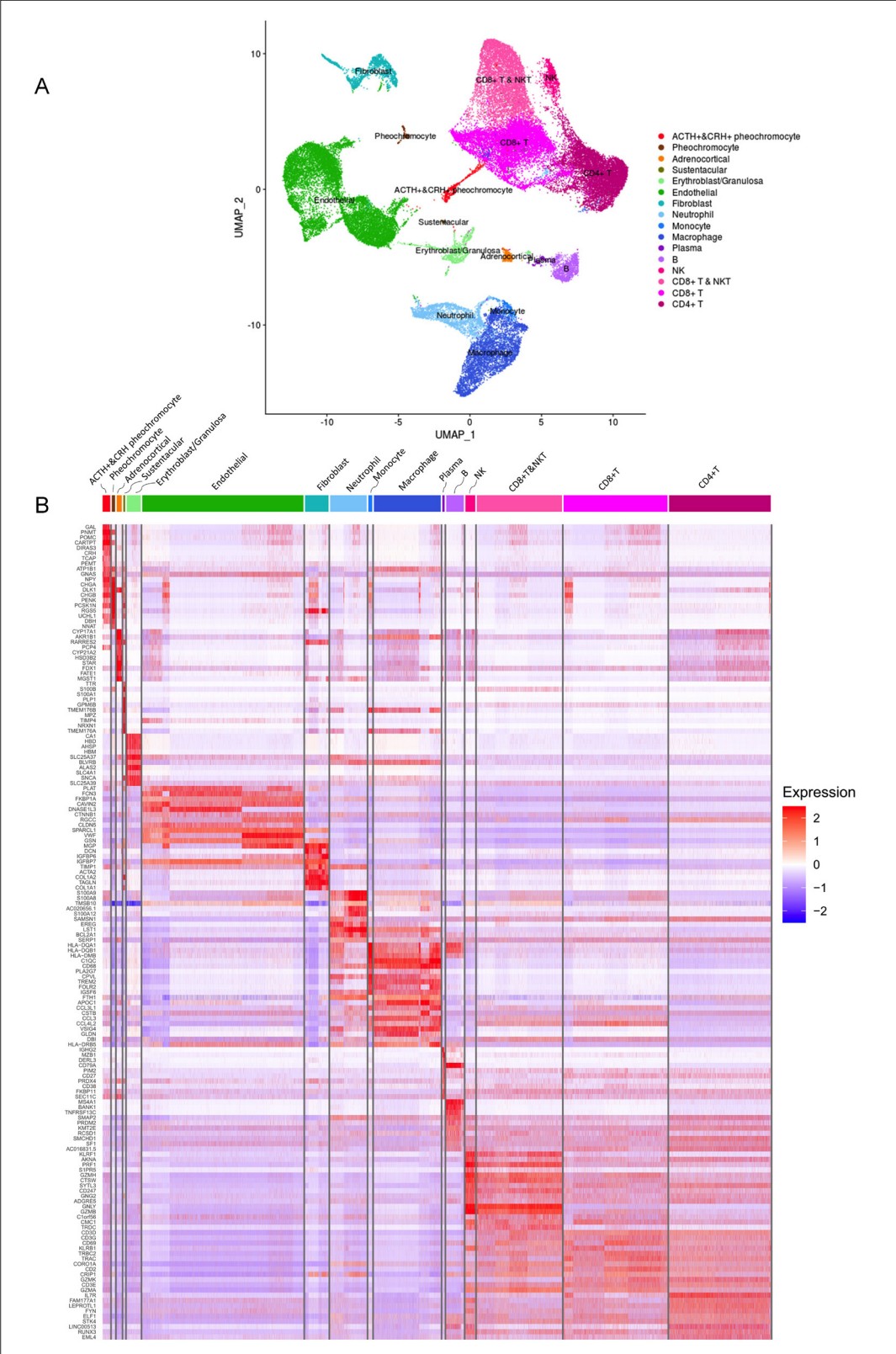

**Figure 2.** Different cell types and their highly expressed genes through single-cell transcriptomic analysis.
(**A**) The t-distributed stochastic neighbor embedding (t-SNE) plot shows 16 main cell types from all specimens.
(**B**) Heatmap shows the scaled expression patterns of the top 10 marker genes in each cell type. The color keys from white to red indicate relative expression levels from low to high.

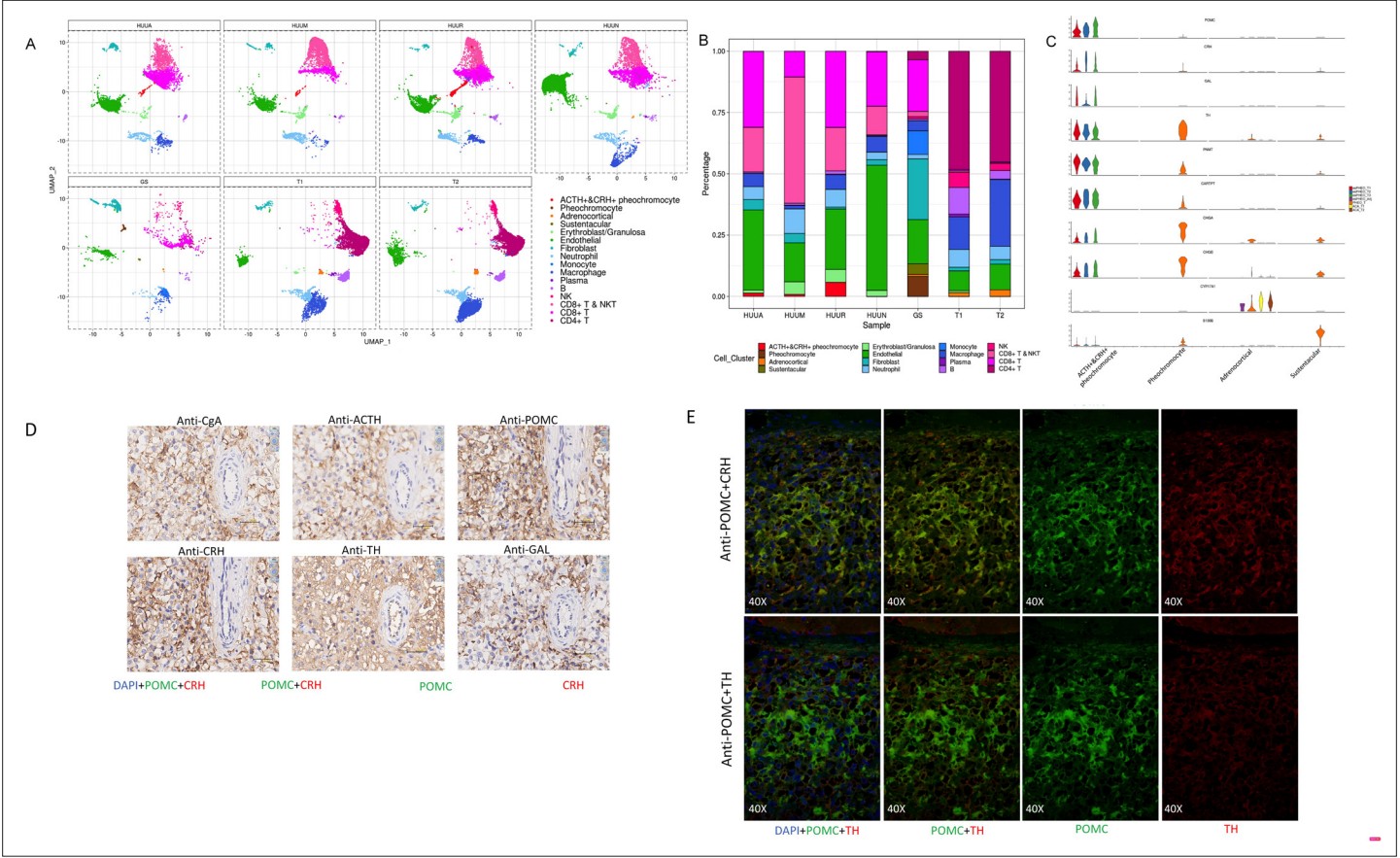

**Figure 3.** A unique tumor cell type was revealed by the composition analysis of cell types in each sample. The results validated an ectopic ACTH and CRH secreting pheochromocytoma. (**A**) Cell clusters shown in UMAP map can be subdivided by different specimens. (**B**) Frequency distribution of cell types among different samples. (**C**) Violin plots showing the expression of representative well-known and potential new marker genes of the four subtypes of adrenal cell types. (**D**) Immunohistochemistry of CgA, ACTH, POMC, CRH, TH, or GAL on serial biopsies from tumor specimen esPHEO_T3. (**E**) Immunofluorescence co-staining for POMC&CRH and POMC&TH on two serial biopsies from tumor specimen esPHEO_T3. UMAP, Uniform Manifold Approximation and Projection.

1. The cell cluster of ACTH+&CRH + pheochromocyte was supported by the specific expression of the markers POMC (proopiomelanocortin) and CRH (corticotropin-releasing hormone) (*Figure 3C*). POMC is a precursor of ACTH, and CRH is the most important regulator of ACTH secretion. We also detected another specific expression signal, GAL, for the cell cluster of ACTH+&CRH + pheochromocyte (*Figure 3C*). GAL encodes small neuroendocrine peptides and can regulate diverse physiologic functions, including growth hormone, insulin release, and adrenal secretion (*Ottlecz et al., 1988*; *McKnight et al., 1992*; *Murakami et al., 1989*; *Hooi et al., 1990*). A study found that GAL and ACTH were co-expressed in human pituitary and pituitary adenomas, and suggested that GAL may be locally involved in the regulation of the HPA axis (*Hsu et al., 1991*). We demonstrated that GAL was expressed in the ACTH+&CRH + pheochromocyte and might participate in the regulation ATCH secretion (*Figure 3C*). Then we examined the known adrenal chromaffin cell markers (CHGA and CHGB) and the markers for catecholamine-synthesizing enzymes (TH and PNMT) (*Figure 3C*). These known markers and another new candidate marker CARTPT were observed in both ACTH+&CRH + pheochromocyte and pheochromocyte cell subtypes. The CYP17A1 and CYP21A2, the typical markers of the adrenal cortical cell subtype, were also investigated (*Figure 3C*). They are members of the cytochrome P450 superfamily, encoding key enzymes, and maybe the precursors of cortisol in the adrenal glucocorticoids biosynthesis pathway (*Auchus et al., 1998*; *Petrunak et al., 2014*). Finally, a subtype of cells with positive expression of S100B was identified, called sustentacular cells. Sustentacular cells were found near chromaffin cells and nerve terminations. Several studies have shown that

sustentacular cells exhibit stem-like characteristics (*Pardal et al., 2007*; *Fitzgerald et al., 2009*; *Poli et al., 2019*; *Scriba et al., 2020*).

Our scRNA-seq analysis validated that the mRNA expression of POMC (precursor for ACTH) and CRH in pheochromocyte triggered the pathophysiology of ectopic ACTH and CRH syndromes, thereby stimulating the adrenal glands to release cortisol. The overexpression of TH and PNMT was responsible for the excessive secretion of catecholamines in the ACTH+&CRH + pheochromocyte and pheochromocyte cell subtypes. Tumor samples (esPHEO_T1, esPHEO_T2, and esPHEO_T3) from Case 1 and PHEO_T from Case 2 were demonstrated to have the function of producing catecholamine. These genes related to catecholamine secretion were all negative for adrenocortical cell subtypes because the catecholamine-producing pheochromocytomas originated from chromaffin cells in the adrenal medulla rather than the adrenal cortex. Our laboratory tests were consistent with these results, that is, both Case 1 and Case 2 had a high level of catecholamines in plasma and 24 hr urine while Case 3 had a normal level. We also found CARTPT was similar to PNMT and can be used as a marker for ACTH+&CRH + pheochromocyte and pheochromocyte. Chromaffin cell markers CHGA and CHGB were mainly characterized in PHEO_T and three tumor samples from Case 1. Adrenocortical cell clusters mainly existed in ACA_T1 and ACA_T2, but a few existed in esPHEO_Adj. S100B was specifically identified in PHEO_T. An absence of S100-positive sustentacular cells has been previously confirmed in most malignant adrenal pheochromocytomas, and the locally aggressive or recurrent group usually contains a large number of these cells (*Unger et al., 1991*). It suggests that PHEO_T from Case 2 might be a locally aggressive case, while Case 1 is the opposite. To validate this finding, we performed additional IHC staining experiments on paraffin-embedded serial slices with similar tissue regions from the tumor specimen esPHEO_T3 using antibodies against CgA, ACTH, POMC, CRH, TH, and GAL. We did find that these markers were all positive in the tumor tissue, which further indicated that the special rare pheochromocytoma exhibited multiple hormone-secreting characteristics, including ACTH, CRH, and catecholamines (*Figure 3D*, *Appendix 1—figure 8*). We also prepared two serial slices for immunofluorescence co-staining for POMC&CRH and POMC&TH. The legible co-localization signals were observed, where the green signal was for POMC, and the red signal was for CRH and TH (*Figure 3E*, *Appendix 1—figure 9*). This result confirmed the ACTH and CRH secreting pheochromocytoma from Case 1 contained a unique multi-functional chromaffin-like cell type, which was consistent with the analysis result by scRNA-seq.

## Differential expression genes show adrenal tumor cell-type specificity

Next, we analyzed the DEGs between ACTH+&CRH + pheochromocyte and the other two subtypes of adrenal tumor cells (pheochromocyte and adrenocortical cells). It is worth noting that many genes were dramatically upregulated specifically in ACTH+&CRH + pheochromocyte when compared with the other tumor cell types, such as GAL, POMC, PNMT, and CARTPT (*Figure 4A*). Using these upregulated or downregulated genes, we performed functional enrichment analysis based on gene ontology (GO) annotation to further characterize the molecular characteristics of different tumor cell types. In comparison with adrenocortical cell types, the highly upregulated genes of ACTH+&CRH + pheochromocyte were mainly enriched in the neuropeptide signaling pathway, hormone secretion, and transport, while the downregulated genes were mostly enriched in the pathway of adrenocortical hormones (*Figure 4B*). Comparing the two types of pheochromocyte, GO functional enrichment analysis for the biology process (BP) revealed that the upregulated genes for ACTH+&CRH + pheochromocyte were also enriched in the neuropeptide signaling pathway, while the enrichment of the downregulated genes from the GO functional result hardly reach statistical significance. Interestingly, compared with adrenocortical cells, a total of 248 upregulated and 198 downregulated genes were detected in ACTH+&CRH + pheochromocyte, while only 95 upregulated and 111 downregulated genes were detected in ACTH+&CRH + pheochromocyte when compared with pheochromocyte (*Figure 4C*), which suggested that the difference between ACTH+&CRH + pheochromocyte and pheochromocyte was relatively small. The known adrenal chromaffin cell markers (CHGA and CHGB) were differential expressed significantly between ACTH+&CRH + pheochromocyte and adrenocortical cells, but not observed significant difference between two subtypes of pheochromocytes. Besides, the co-upregulated genes, such as CARTPT, PNMT, POMC, GAL, and CRH, were responsible for the production of a variety of hormones and involved in neuropeptide signaling pathways. Of which, the product of PNMT catalyzes the last step of the catecholamine biosynthesis pathway, methylating norepinephrine to form

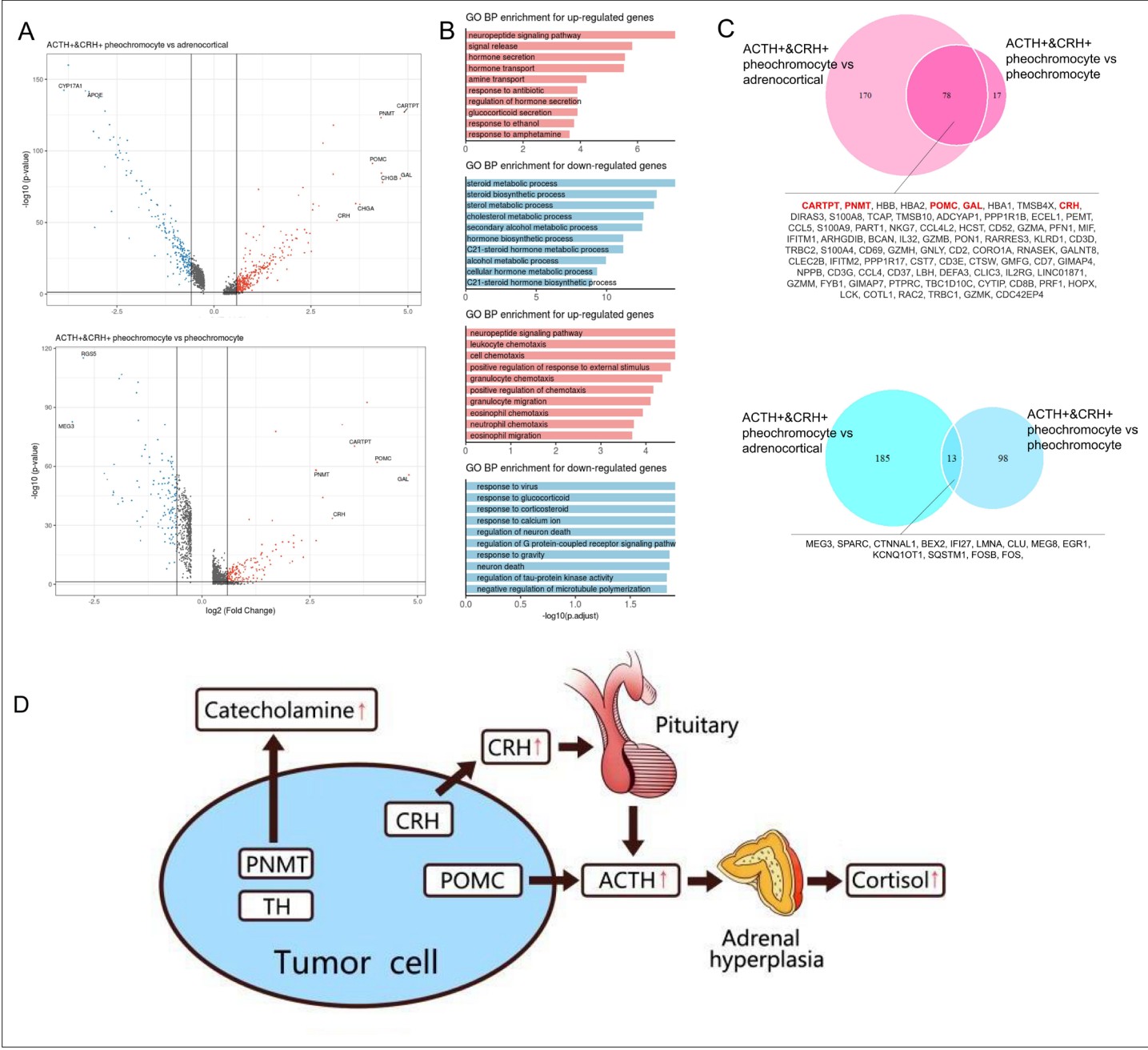

**Figure 4.** Altered functions in POMC+&CRH + pheochromocyte revealed by differential gene expression analysis. (**A**) Volcano plot of changes in gene expression between POMC+&CRH + pheochromocytes and other adrenal cell types (pheochromocytes and adrenocortical cells). The x-axis specifies the natural logarithm of the fold-changes (FC) and the y-axis specifies the negative logarithm of the adjusted p-values at the base of 10. Gray vertical and horizontal lines reflect the filtering criteria. Red and blue dots represent the genes with significantly higher expression and lower expression, respectively. (**B**) Functional enrichment analysis of the upregulated or downregulated gene revealed the top altered terms in the biological process of gene ontology. The x-axis specifies the negative logarithm of the adjusted p-values at the base of 10. (**C**) Overlap of upregulated or downregulated genes in the two pairwise comparisons between adrenal cell types. The red font highlights the well-known and potential markers that have been identified. (**D**) The molecular mechanism of CS in the rare case of ectopic ACTH and CRH syndrome is associated with pheochromocytoma. The pheochromocytoma tumor of Case 1 is composed of a unique chromaffin-like cell type that directly expresses both ACTH and CRH; the expression of POMC directly causes the secretion of ACTH, and the expression of CRH indirectly promotes the secretion of ACTH hormone, which ultimately leads to CS.

epinephrine. The overexpression of PNMT was responsible for the significantly elevated epinephrine (*Appendix 1—table 1*) of the rare Case 1 with ectopic ACTH and CRH secretory pheochromocytoma. The elevated plasma ACTH (*Appendix 1—table 1*) of the rare Case 1 could be explained by specific high expression signals of GAL, POMC, and CRH. In details, POMC is the precursor of ACTH; CRH is the most important regulator of ACTH secretion; and GAL was co-expressed in the ACTH+&CRH + pheochromocyte, which might be locally involved in the regulation of the HPA axis. Therefore, we concluded that the tumor cell type of ACTH+&CRH + pheochromocyte from Case 1 had multiple hormone secretion functions, namely, CRH secretion function, ACTH secretion function, and catechol-amine secretion function. Furthermore, we believed that the rare Case 1 harbor the ACTH-dependent CS is due to the presence of the identified novel tumor cell type of ACTH+&CRH + pheochromocyte, which secretes both ACTH and CRH. The secretion of ACTH had a direct effect on the adrenal gland to produce cortisol, while the secretion of CRH can indirectly stimulate the secretion of ACTH from the anterior pituitary (*Figure 4D*).

## RNA velocity analysis

To investigate dynamic information in individual cells, we performed RNA velocity analysis using velo-cyto.py for spliced or unspliced transcripts annotation followed by scVelo pipeline for RNA dynamics modeling. RNA velocity is the time derivative of the measured mRNA abundance (spliced/unspliced transcripts) and allows to estimate the future developmental directionality of each cell (*La Manno et al., 2018*). We observed the ratios of spliced and unspliced mRNA, and sustentacular cell type was ranking first with 36% unspliced proportions among non-immune cell types (*Figure 5A and B*). The balance of unspliced and spliced mRNA abundance is an indicator of the future state of mature mRNA abundance, and thus the future state of the cell (*Bergen et al., 2020*). Previously study had observed unspliced transcripts were enriched in genes involved in DNA binding and RNA processing in hematopoietic stem cells (*Bowman et al., 2006*). For the high proportions of unspliced/spliced transcripts, stem-like characteristics of sustentacular cells were supported. There were more spliced transcripts proportions in POMC+&CRH + pheochromocytes than in pheochromocytes (*Figure 5B*). Then, we estimated pseudotime grounded on transcriptional dynamics and generated velocity streamlines that account for speed and direction of motion. As observed in the pseudotime of four adrenal cell subtypes, medullary cells are earlier than cortical cells (*Figure 5C*). From velocity stream-lines, we found the four adrenal cell subtypes, that is, POMC+&CRH + pheochromocytes, pheochro-mocytes adrenocortical cells, and sustentacular cells, were independent respectively and not directed toward other cell types (*Figure 5D*). Newly transcribed, unspliced pre-mRNAs were distinguished from mature, spliced mRNAs by detecting the presence of introns. Genes, like POMC and CRH, only contain one coding sequence (CDS) region, were all detected as spliced (*Appendix 1—figure 5*). It indicated that the actual values of RNA velocity for POMC+&CRH + pheochromocytes might be larger than the predicted ones. Furthermore, the spliced versus unspliced phase for CHGA, CHGB, and TH demonstrated a clear more dynamics expression in POMC+&CRH + pheochromocytes than in pheochromocytes (*Appendix 1—figure 5*).

## Lineage tracing analysis confirms the plasticity of adrenal tumor cell subsets

We performed the pseudotime analysis for the adrenal tumor cell subsets to determine the pattern of the dynamic cell transitional states. We used the recommended strategy of Monocle to order cells based on genes that differ between clusters. The sustentacular cells were in an early state in pseudotime analysis (*Figure 6A, B and C*), which was in accordance with their exhibited stem-like properties and the highest unspliced proportion among non-immune cell types in the RNA velocity analysis. The results also showed a transition from sustentacular cells to pheochromocytes and then to ACTH+&CRH + pheochromocyte, and adrenocortical cells were on another branch (*Figure 6A, B and C*). To determine whether specific gene modules might be responsible for this cell plasticity, we calculated the expression levels of all the genes in the single-cell transcriptome identified the DEGs on the different paths through the entire trajectory (*Figure 6D*), which showed the dynamic changes of each gene over pseudotime.

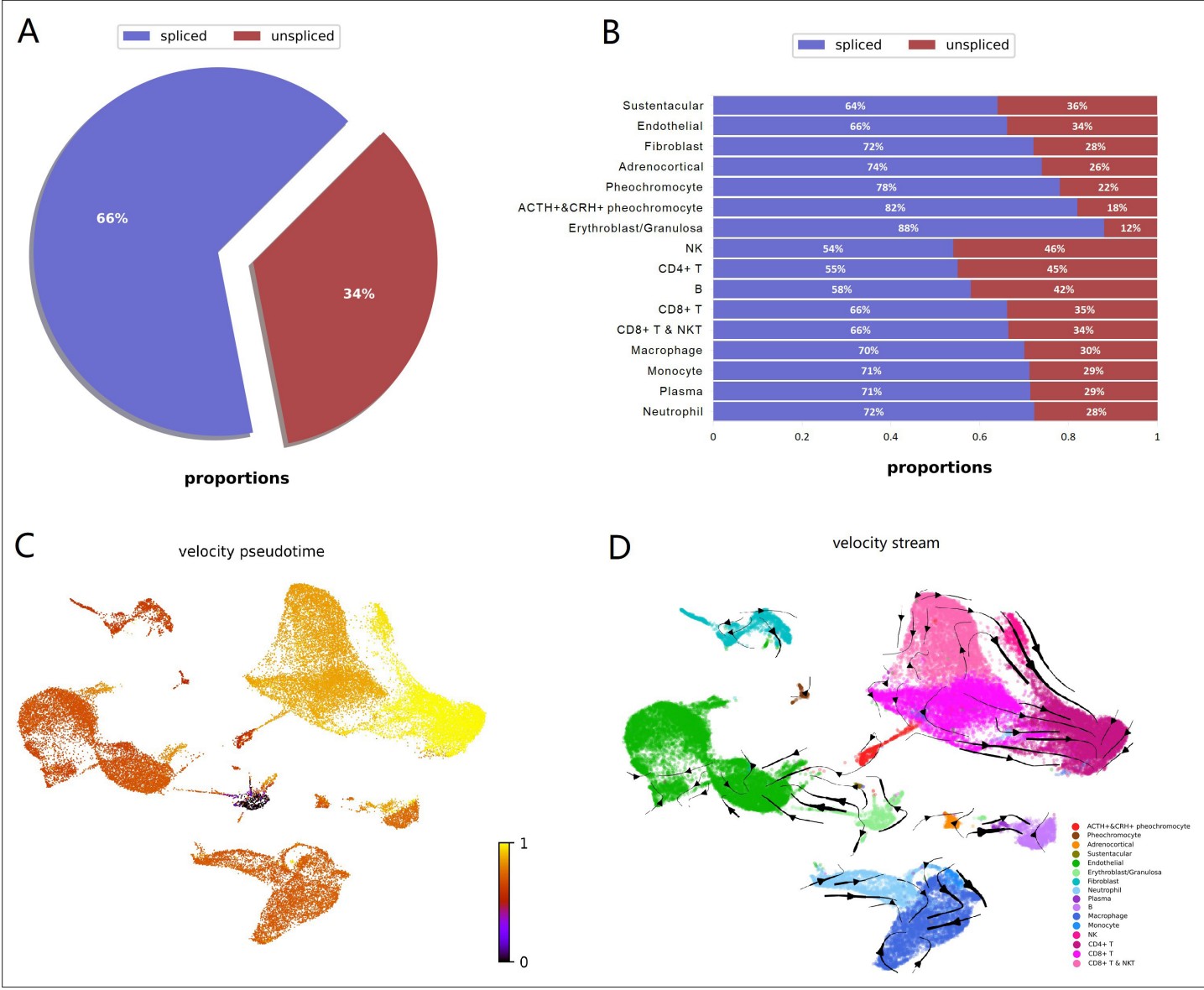

**Figure 5.** RNA velocity analysis supported sustentacular cells as root and indicated four adrenal cell subtypes were independent respectively and not directed toward other cell types. RNA velocity is the time derivative of the measured mRNA abundance (spliced/unspliced transcripts) and allows to estimate the future developmental directionality of each cell. (**A**) The total ratios of spliced and unspliced transcripts. (**B**) The ratios of spliced and unspliced transcripts for each cell cluster. Sustentacular cells exhibit high proportions of unspliced/spliced transcripts, which support the stem-like characteristics. (**C**) The pseudotime, derived from the estimation of RNA velocity, is visualized in UMAP plot. It indicated that medullary cells were earlier than cortical cells. (**D**) The direction of cell differentiation inferred from estimated RNA velocities is plotted as streamlines on the UMAP. No consistent velocity direction was observed for the four adrenal cell subtypes, that is, POMC+&CRH + pheochromocytes, pheochromocytes adrenocortical cells, and sustentacular cells. UMAP, Uniform Manifold Approximation and Projection.

## scRNA-seq reveals distinct immune and endothelial cell type in the tumor microenvironment

scRNA-seq allowed us to use an unbiased approach to discover the composition of immune cell populations of the adrenal tumor specimens. Analysis of our transcriptional profiles revealed that from the frequency distribution of cell clusters, immune cells accounted for more than ~50% of total cells (*Figure 3B*). We identified and annotated the immune cell types based on the expression of conventional markers, such as B cells with MS4A1, NK cells with GNLY, and Neutrophil with S100A8 and S100A9 (*Figure 7A*). The various frequency distribution of immune cell sub-clusters was observed among different samples (*Figure 7B*). Due to the identical tumor microenvironment, all three tumor

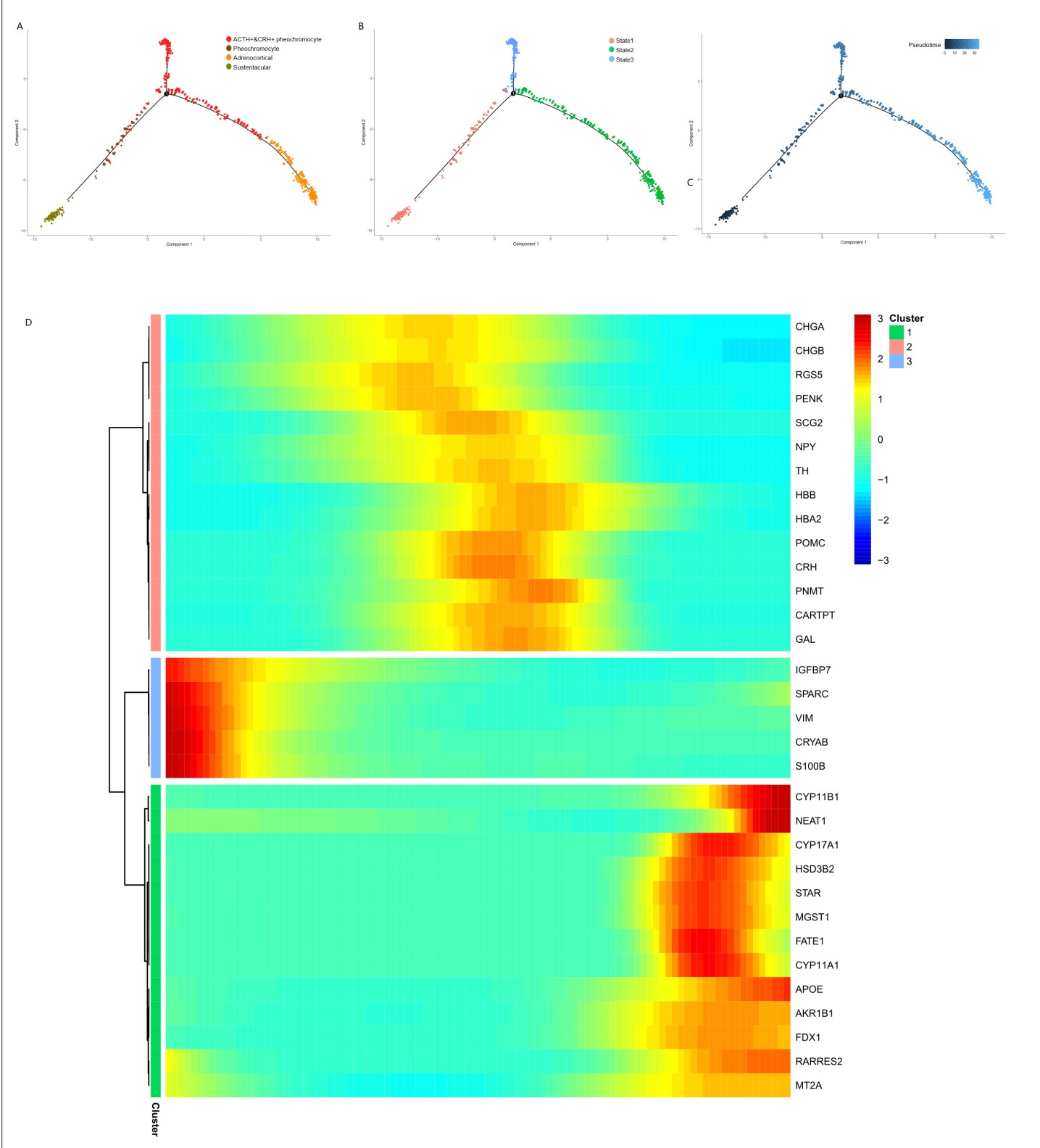

**Figure 6.** Pseudotime analysis of adrenal cells inferred by Monocle. We ran reduce dimension with t-SNE for four types of adrenal cells and sorted cells along pseudotime using Monocle. The single-cell pseudotime trajectories by ordering cells were constructed based on genes that differ between clusters. Each point corresponds to a cell. (**A**) Cells are colored in each adrenal cell type. (**B**) Cells colors indicated the cell's stage in simulated transition. (**C**) Cells colored by pseudotime. (**D**) Heatmap of gene markers that are significantly changed in cell-type transition in pseudotime. Each row represents

*Figure 6 continued on next page*

*Figure 6 continued*

a gene, where the left end corresponds to the transition starting point. (sustentacular cells) and the right end corresponds to transition ending point (adrenocortical cells). Color scheme represents the z-score distribution from 3.0 (blue) to 3.0 (red). Genes that covary across transition are clustered into three blocks.

specimens one peritumoral specimen from the rare case had similar immune cell composition. Interestingly, the CD4 T cells, B cells, and macrophages are mainly presented in two adrenal cortical adenomas (ACA_T1 and ACA_T2), while the CD8 T cells mostly resided in the microenvironment of other pheochromocytoma tumor and the peritumoral specimen. We found the heterogeneity of T cells in different adrenal tumor subtypes, that is, compared with CD4 T cells in adrenocortical adenomas, the pheochromocytoma types were mostly manifested by activated CD8+, especially in the anatomic specimens from the ectopic ACTH&CRH secreting pheochromocytoma.

Endothelial cells consisted of four distinct sub-clusters: vascular endothelial cells, lymphatic endothelial cells, cortical endothelial cells, and other endothelial cells, as shown in the cell cluster distribution map highlighted by endothelial cells (*Figure 8A*, *Supplementary file 3*). Various adrenal tumor subtypes had different endothelial compositions (*Figure 8B*). Vascular endothelial cells were mainly identified in pheochromocytoma samples (esPHEO_T1, esPHEO_T2, esPHEO_T3, and PHEO_T), because pheochromocytoma is a tumor arising in the adrenal medulla, and vascular endothelial cells might be detected from the medullary capillary. Cortical endothelial cells were mainly detected in adrenocortical adenomas (ACA_T1 and ACA_T2). Lymphatic endothelial cells were found in the adjacent adrenal specimen of the rare ACTH+&CRH + pheochromocytoma (esPHEO_Adj). Then, by comparing vascular endothelial cells with two other subclusters (lymphatic endothelial cells and cortical endothelial cells), we found the markers across the subclusters of endothelial cells and annotated GO function of differentially expressed genes (*Figure 8C and D*). Vascular endothelial cells are the barrier between the blood and vascular wall and have the functions of organizing the extracellular matrix and regulating the metabolism of vasoactive substances. Lymphatic endothelial cells are responsible for chemokine-mediated pathways. Cortical endothelial cells express TFF3 and FABP4, which are involved in repairing and maintaining stable functions.

## Discussion

Both CS and pheochromocytoma are serious clinical conditions. In this study, we reported an extremely rare patient (Case 1) with ATCH-dependent CS due to an ectopic ACTH&CRH secreting pheochromocytoma. Surgery is the most common treatment strategy for this type of tumor. After the operation, our clinical manifestations of Case 1 showed the complete remission of CS. The IHC of the dissected tumor confirmed the diagnosis with positive staining for CRH and ACTH. In this study, scRNA-seq was used for the first time to identify the rare ACTH+&CRH + pheochromocyte cell subset. Compared with other subtypes of adrenal tumors, the common pheochromocytoma (from Case 2) and adrenal cortical cells (from Case 3), the DEGs in Case 1 were further characterized. Case 2 was examined to have normal levels of cortisol and ACTH, but Case 3 showed a Cushingoid appearance. The molecular mechanism of CS in Case 3 was different, which was attributed to two cortical adenomas on the left adrenal, showing ACTH-independent hypercortisolemia. In addition, to investigate the genetic driver for Case 1, we supplemented whole-exome sequencing experiments for all rest specimens, that is, tumors (esPHEO_T2 and esPHEO_T3) and controls (esPHEO_Adj and esPHEO_Blood) from the rare case with ectopic ACTH&CRH-secreting pheochromocytoma. Filtered germline and somatic mutations were listed in *Supplementary file 4* including detailed annotations. Genetic mutations of phaeochromocytoma and paraganglioma are mainly classified into two major clusters, that is, pseudo hypoxic pathway and kinase signaling pathways (*Pillai et al., 2016*; *Nölting and Grossman, 2012*). We did not find any gene mutations that were related to these two major clusters. We only identified one shared somatic variant of *ACAN* (c.5951T > A:p.L1984Q) comparing variants in tumor samples to controls but Sanger sequencing only confirmed the presence in esPHEO_T3 which was not observed in esPHEO_T2 (*Appendix 1—figure 7*). *ACAN*, encoding a major component of the extracellular matrix, is a member of the aggrecan/versican proteoglycan family. Mutations of *ACAN* were reported related to steroid levels (*Yousri et al., 2018*). It is well-established that circulating steroid levels are linked to inflammation diseases such as arthritis, because arthritis as well as most autoimmune disorders

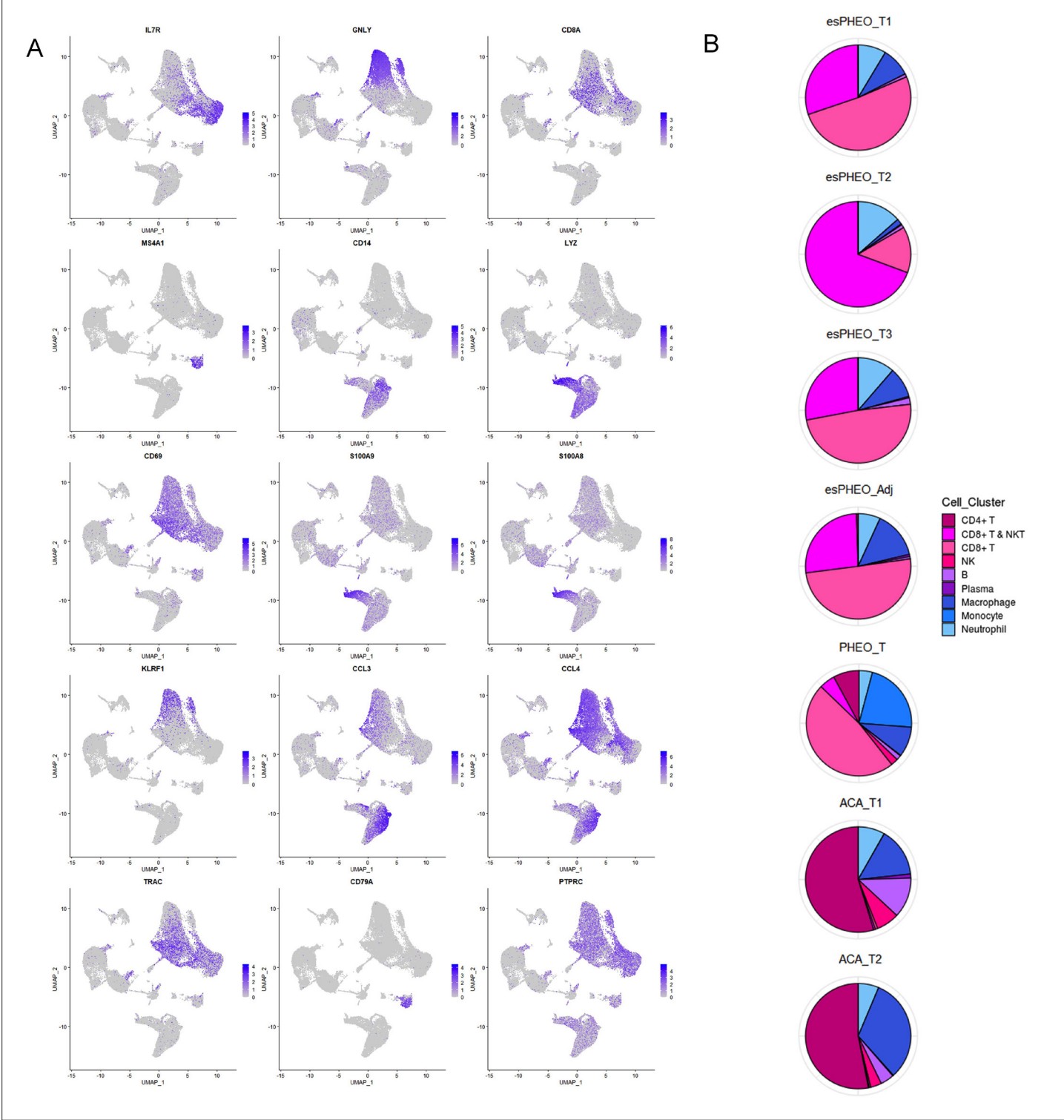

**Figure 7.** Diverse immune microenvironments in different adrenal tumor subtypes and tumor-adjacent tissue. (**A**) The UMAP diagram shows the expression levels of well-known marker genes of immune cell types. (**B**) Frequency distribution of immune cell sub-clusters in different adrenal tumors and tumor-adjacent tissue. UMAP, Uniform Manifold Approximation and Projection.

results from a combination of several predisposing factors including the stress response system such as hypothalamic-pituitary-adrenocortical axis (*Cutolo et al., 2003*). But no direct evidence related to *ACAN* to phaeochromocytoma. Therefore, no obvious genetic driver was found to explain the

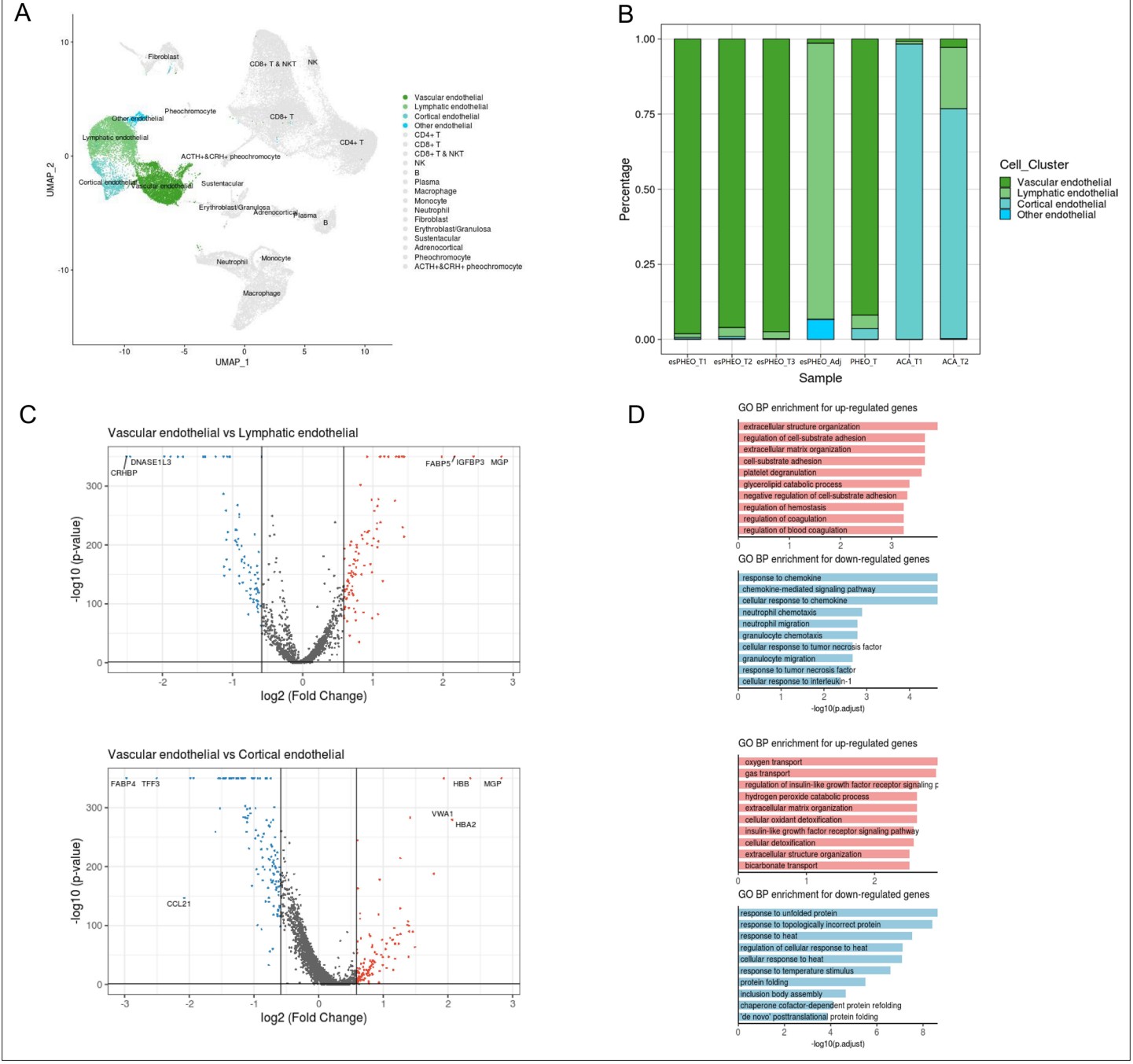

**Figure 8.** Differential gene expression analysis shows changes in endothelial cell functions. (**A**) The UMAP diagram shows four different endothelial cell sub-clusters. (**B**) Frequency distribution of endothelial cell sub-clusters among different adrenal tumors and tumor-adjacent specimen. (**C**) Volcano plot of changes in gene expression for the endothelial cluster by comparing vascular endothelial cells with lymphatic cells and cortical endothelial cells, respectively. The x-axis specifies the natural logarithm of fold-changes (FC) and the y-axis specifies the negative logarithm of the adjusted p-values at the base of 10. Gray vertical and horizontal lines reflect the filtering criteria. Red and blue dots represent the genes with significantly higher expression and lower expression, respectively. (**D**) Functional enrichment analysis of the upregulated or downregulated gene revealed top altered functional terms. The x-axis specifies the negative logarithm of the adjusted p-values at the base of 10. UMAP, Uniform Manifold Approximation and Projection.

rare case of ACTH/CRH-secreting phaeochromocytoma. Further investigations would be needed to uncover the relation between *ACAN* and phaeochromocytoma.

For many years, the understanding of neurotransmission has been dominated by the concept of 'one cell, one hormone, and one neuron one transmitter,' which is known as Dale's Principle (Dale in

1934; for detailed discussion, see *Burnstock, 1976*; *Burnstock, 1976*). *Sakuma et al., 2016* reported an ectopic ACTH pheochromocytoma case and proved that ACTH and catecholamine were produced by two functionally distinct chromaffin-like tumor cell types through immunohistochemical analysis *Sakuma et al., 2016*. However, more and more evidence has emerged that Dale's principle is incorrect because existing studies have shown that these cells are multi-messenger systems (*Hakanson and Sundler, 1983*; *Apergis-Schoute et al., 2019*; *Svensson et al., 2018*). Based on scRNA-seq results, we concluded that the tumor cells from Case 1 had multiple hormone secretion functions, namely, CRH secretion function, ACTH secretion function, and catecholamine secretion function. CRH is the most important regulator of ACTH secretion. Therefore, we believed that the secretion of both CRH and ACTH of this tumor led to ACTH-dependent CS. Besides, the secretion of ACTH had a direct impact on the adrenal gland to produce cortisol, and the secretion of CRH indirectly stimulated the secretion of ACTH by the anterior pituitary. *Jessop et al., 1987* also draw the same conclusion in their report in 1987. However, in the reported case, the histological immunostained result was shown only for the corticotropin-releasing factor (CRF-41), but not for ACTH (*Jessop et al., 1987*).

Adrenal glands are composed of two main tissue types, namely, the cortex and the medulla, which are responsible for producing steroid and catecholamine hormones, respectively. The inner medulla is derived from neuroectodermal cells of neural crest origin, while the outer cortex is derived from the intermediate mesoderm. In the adrenal pheochromocytomas, a third cell type with the positive expression of S100B was identified, called 'sustentacular' cells (*Suzuki and Kachi, 1995*; *Lloyd et al., 1985*). By evaluating 17 malignant and recurrent or locally aggressive adrenal pheochromocytomas, *Unger et al., 1991* found that sustentacular cells were absent in most malignant cases (*Unger et al., 1991*). Because there are no sustentacular cells in ACTH&CRH secreting pheochromocytoma, ACTH&CRH secreting pheochromocytoma is more serious than the common pheochromocytoma. Furthermore, several studies have demonstrated that sustentacular cells exhibit stem-like characteristics (*Pardal et al., 2007*; *Fitzgerald et al., 2009*; *Poli et al., 2019*; *Scriba et al., 2020*). A unique case of a tumor originating from S100-positive sustentacular cells was previously reported (*Lau et al., 2006*). The RNA velocity estimation and pseudo-time analysis of different adrenal cell subtypes supported the sustentacular cells exhibiting stem-like properties. Although pheochromocyte was prior to ACTH&CRH secreting pheochromocyte in pseudotime order, the RNA velocity prediction of POMC+&CRH+ pheochromocytes might be under-estimated because the transcripts of POMC and CRH were all predicted as spliced ones. Based on the spliced versus unspliced phase for CHGA, CHGB, and TH, it showed a clear more dynamics expression in POMC+&CRH+ pheochromocytes than in pheochromocytes. We assumed that ACTH&CRH secreting pheochromocyte have more hormone-producing functions, retain stem- and endocrine-differentiation ability. But further experiments are needed to validate our hypothesis.

There are bidirectional communications between the immune system and the neuroendocrine system (*Blalock, 1989*). Hormones produced in the endocrine system, especially glucocorticoids, affect the immune system to modulate its function (*Imura and Fukata, 1994*). Other hormones, such as growth hormone (GH) and prolactin (PRL), also modulate the immune system (*Blalock, 1989*). It has been proved that the exogenous production of cytokines can stimulate and mediate the release of multiple hormones including ACTH, CRH (*Rivier et al., 1989*; *Bernton et al., 1987*), and induce the activation of the HPA axis (*Gisslinger et al., 1993*; *Fukata et al., 1994*; *Kakucska et al., 1993*; *Murakami N Fukata et al., 1992*). Human T cells coordinate the adaptive immunity of different anatomic compartments by producing cytokines and effector molecules (*Szabo et al., 2019*). The activation of naive T cells through the antigen-specific T cell receptor (TCR) can initiate transcriptional programs that can drive the differentiation of lineage-specific effector functions. CD4+ T cells secrete cytokines to recruit and activate other immune cells, while CD8+ T cells have cytotoxic functions and can directly kill infected or tumor cells. Recent studies have shown that the composition of the T cell subset is related to the specific tissue locations (*Carpenter et al., 2018*; *Thome et al., 2014*). scRNA-seq can be used to deconvolve the immune system heterogeneity with high resolution. Compared with adrenocortical adenomas which were in CD4+ (with the expression of cytokine receptors, such as the IL-7R) state, T cells in pheochromocytoma, especially T cells in the ectopic ACTH&CRH secreting pheochromocytoma were inactivated CD8+ state, suggesting different tumor microenvironments between adrenocortical adenomas and pheochromocytoma. Previous studies have shown that signaling through IL-7R is essential in the developmental process and regulation of

lymphoid cells (*Kondrack et al., 2003*; *Tan et al., 2001*; *Tan et al., 2002*; *Lenz et al., 2004*; *Li et al., 2003*; *Seddon et al., 2003*), and disruption of the IL-7R signaling pathway may lead to skewed T cell distribution and cause immunodeficiency (*Maraskovsky et al., 1996*; *Kaech et al., 2003*; *Carini et al., 1994*). Our results indicated the heterogeneity of the immune system between different samples, and CD4+ T cells with the high expression level of IL-7R might be related to adrenal tumor progression, apoptosis, or factors influencing progression such as immune activation. Although we have shown the heterogeneity of immune cell types in different adrenal tumor subtypes, it is unclear how T cells influence different markers, including effector states and interferon-response states. In addition to composition differences, a deeper understanding of the complex interactions between adrenal tumor tissues and immune systems is a key issue in neuroendocrine tumor research.

Overall, we reported a rare case in which ectopic ACTH&CRH-secreting pheochromocytoma on the left adrenal that infiltrated around the kidney and psoas major tissues. We applied scRNA-seq to identify this rare and special adrenal tumor cell. Thus, the majority of our analysis focused on the validation of novel tumor cell type and their multiple hormones-secreting functions, namely, CRH secretion function, ACTH secretion function, and catecholamine secretion function. Also, GAL could be a candidate marker to detect the rare ectopic ACTH+&CRH + secreting pheochromocytes. For future studies, on one hand, we are very concerned about similar suspicious cases in the clinic. On the other hand, we are going for following research for further downstream experiments to validate the molecular mechanism for secreting multiple hormones.

# Materials and methods

## Clinical specimens collection

Our study included three adrenal tumor patients, that is, pheochromocytoma with ectopic ACTH and CRH secretion, common pheochromocytoma, and adrenocortical adenoma. All three patients had signed the consent forms at the General Surgery Department of Peking Union Medical College Hospital (PUMCH). The enhanced CT scanning images and laboratory test (ACTH, 24 hr urine-free cortisol, Catecholamines) of relevant patients are listed in Appendix 1. Fresh tumor specimens were collected during surgical resection. For the case of ACTH and CRH secreting pheochromocytoma, we performed the surgical resection of the tumor at left adrenal (esPHEO_T1) and its infiltrating tissues located in the kidney (esPHEO_T3) and masses (esPHEO_T2), and obtained three tumor specimens. The peritumor sample (esPHEO_Adj) was collected from the left adrenal tissue under the supervision of a qualified pathologist. The other two patients underwent left adrenalectomy and provided the other three tumor specimens. In details, one tumor specimen was obtained from the patient with common pheochromocytoma and two tumor specimens were obtained from the patient with adrenocortical adenoma. A total of seven specimens were carefully dissected under the microscope and confirmed by a qualified pathologist.

## Single-cell transcriptome library preparation and sequencing

After the resection, tissue specimens were rapidly processed for single-cell RNA sequencing.

Single-cell suspensions were prepared according to the protocol of Chromium Single Cell 3' Solution (V2 chemistry). All specimens were washed two times with cold 1× phosphate-buffered saline (PBS). Haemocytometer (Thermo Fisher Scientific) was used to evaluate cell viability rates. Then, we used Countess (Thermo Fisher Scientific) to count the concentration of single-cell suspension, and adjust the concentration to 1000 cells/µl. Samples that were lower than the required cell concentration defined in the user guide (i.e., <400 cells/µl) were pelleted and re-suspended in a reduced volume; and then the concentration of the new solution was counted again. Finally, the cells of the sample were loaded, and the libraries were constructed using a Chromium Single-Cell Kit (version 2). Single-cell libraries were submitted to 150 bp paired-end sequencing on the Illumina NavoSeq platform.

## Single-cell RNA-seq data pre-processing and quality control

After obtaining the paired-end raw reads, we used CellRanger (10× Genomics, v3.1.0) to pre-process the single-cell RNA-seq data. Cell barcodes and unique molecular identifiers (UMIs) of the library were extracted from read1. Then, the reads were split according to their cell (barcode) IDs, and the UMI

sequences from read2 were simultaneously recorded for each cell. Quality control on these raw readings was subsequently performed to eliminate adapter contamination, duplicates, and low-quality bases. After filtering barcodes and low-quality readings that were not related to cells, we used STAR (version 2.5.1b) to map the cleaned readings to the human genome (hg19) and retained the uniquely mapped readings for UMIs counts. Next, we estimated the accurate molecular counts and generated a UMI count matrix for each cell by counting UMIs for each sample. Finally, we generated a gene-barcode matrix that showed the barcoded cells and gene expression counts.

Based on the number of total reads, the number of detected gene features, and the percentage of mitochondrial genes, we performed quality control filtering through Seurat (v3.1.5) (*Butler et al., 2018*; *Stuart et al., 2019*) to discard low-quality cells. Briefly, mitochondrial genes inside one cell were calculated lower than 20%, and total reads in one cell were below 40,000. Also, the cells were further filtered according to the following criteria: PHEO, ACA, and esPHEO samples with no more than 5000, 3000, and 2500 genes were detected, respectively, and at least 200 genes were detected per cell in any sample. Low-quality cells and outliers were discarded, and the single cells that passed the QC criteria were used for downstream analyses. Doublets were predicted by DoubletFinder (v2.0) (*McGinnis et al., 2019*) and DoubletDecon (v1.1.6) (*DePasquale et al., 2019*; *Appendix 1—figure 2*).

## Clustering analysis and cell phenotype recognition

Seurat (*Butler et al., 2018*; *Stuart et al., 2019*) software package was used to perform cell clustering analysis to identify major cell types. All Seurat objects constructed from the filtered UMI-based gene expression matrixes of given samples were merged. We first applied 'SCTransform' function to implement normalization, variance stabilization, and feature selection through a regularized negative binomial model. Then, we reduced dimensionality through PCA. According to standard steps implemented in Seurat, highly variable numbers of principal components (PCs) 1–20 were selected and used for clustering using the t-distributed stochastic neighbor embedding method (t-SNE). We identified cell types of these groups based on the expression of canonic cell type markers or inferred by CellMarker database (*Zhang et al., 2019*). Finally, the four groups of endothelial cells were combined to a larger endothelial cell cluster for downstream analysis. Cellular cluster statistics were added in *Supplementary file 2*, which presented cell counts for each cellular cluster in different samples and top 10 gene markers. Endothelial cell cluster statistics were added in *Supplementary file 3*, which presented cell counts for each endothelial cell cluster in different samples and top 10 gene markers.

## DEG analysis

The cell-type-specific genes were identified by running Seurat (*Butler et al., 2018*; *Stuart et al., 2019*) containing the function of 'FindAllMarkers' on a log-transformed expression matrix with the following parameter settings: min.pct=0.25, logfc.threshold=0.25 (i.e., there is at least 0.25 log-scale fold change between the cells inside and outside a cluster), and only.pos=TRUE (i.e., only positive markers are returned). For heatmap and violin plots, the SCT-transformed data from Seurat pipeline were used. Using the Seurat 'FindMarkers' function, we found the DEGs between two cell types. We also used R package of clusterProfiler with default parameters to identify gene sets that exhibited significant and consistent differences between two given biological states.

## RNA velocity estimation

We used the velocyto python package (v0.17.17) (*La Manno et al., 2018*) for distinguishing transcripts as spliced or unspliced mRNAs based on the presence or absence of intronic regions in the transcript. We took aligned reads of BAM file for each sample as input. After per sample abundance estimation, it generated a LOOM file with the loompy package. Then, we used the scVelo (v0.2.3; *Bergen et al., 2020*) to combine each sample abundance data as well as cell cluster information from the Seurat object. We showed the proportions of abundances for each sample using scvelo.pl.proportions function. The RNA velocity was estimated for each cell for an individual gene at a given time point based on the ratio of its spliced and unspliced transcript. RNA velocity graph was visualized on a UMAP plot, with vector fields representing the averaged velocity of nearby cells. We also visualized some marker genes dynamics portraits with scv.pl.velocity to examine their spliced versus unspliced phase in different cell types.

## Pseudotime analysis

The Monocle2 packages (v2.14.0) (*Trapnell et al., 2014*) for R were used to determine the pseudo-times of the differentiation of four different cell subtypes, that is, POMC+/CRH + pheochromocytoma, pheochromocytoma, adrenocortical, and sustentacular cells. We converted a Seurat3 integrated object into a Monocle cds object and distributed the composed cell clusters to the Monocle cds partitions. Then, we used Monocle2 to perform trajectory graph learning and pseudotemporal sorting analysis by specifying the sustentacular cells as the root nodes. To identify genes that are significantly regulated as the cells differentiate along the cell-to-cell distance trajectory, we used the differentialGeneTest() function implemented in Monocle2 (*Trapnell et al., 2014*). Finally, we selected the genes that were differentially expressed on different paths through the trajectory and plotted the pseudotime_heatmap.

## Gene regulatory network (regulon) analysis

We used R package SCENIC (v1.1.2) (*Aibar et al., 2017*) for gene regulatory network inference. Normalized log counts were used as input to identify co-expression modules by the GRNBoost2 algorithm. Following which, regulons were derived by identifying the direct-binding TF target genes while pruning others based on motif enrichment around transcription start site (TSS) with cisTarget databases. Using aucell, the regulon activity score was measured as the area under the recovery curve (AUC). Additionally, regulon specificity score (RSS) was used for the detection of the cell-type-specific regulons.

## Cell-cell communication analysis

Given the diverse immune and endothelial cell types in the tumor microenvironment, we performed cell-cell communication analysis using CellPhoneDB Python package (2.1.7) (*Efremova et al., 2020*). We visualized the potential cell-cell interactions among various immune cells, endothelial cells, and other cell types in the different tumor microenvironment (esPHEO, esPHEO_Adj, PHEO, and ACA) (*Appendix 1—figure 6*).

## Whole-exome sequencing

Genomic DNA extracted from whole blood (esPHEO_Blood), esPHEO_T2, esPHEO_T3, and esPHEO_Adj of the rare Case 1 were sent for whole-exome sequencing. The exomes were captured using the Agilent SureSelect Human All Exon V6 Kit and the enriched exome libraries were constructed and sequenced on the Illumina NovaSeq 6000 platform to generate WES data (150 bp paired-end reads, >100×) according to standard manufacturer protocols. The cleaned reads were aligned to the human reference genome sequence NCBI Build 38 (hg38) using Burrows-Wheeler Aligner (BWA) (v0.7.17) (*Li and Durbin, 2009*). All aligned BAM were then performed through the same bioinformatics pipeline according to GATK Best Practices (v4.2) (*McKenna et al., 2010*). We obtained germline variants shared by all tumors and control samples based on variant calling from GATK-HaplotypeCaller. We then used GATK-MuTect2 to call somatic variants in tumors and obtained a high-confidence mutation set after rigorous filtering by GATK-FilterMutectCalls. All variants were annotated using ANNOVAR (v2018Apr16) (*Wang et al., 2010*). The criteria for filtering variants were as follows: (1) only retained variants located on exon or splice site, and excluded synonymous variants; (2) retained rare variants with minor allele frequencies <5% in any ancestry population groups from public databases (1000 Genomes, ESP6500, ExAC, or the GnomAD); (3) For germline variants, excluded common variants in dbSNP (Build 138) and predicted benign missense variants by SIFT, Polyphen2, and Mutation Taster.

## Immunocytochemistry and Immunofluorescence

Immunocytochemical and immunofluorescent staining experiments were conducted according to standard protocols using antibodies against malinfixed paraffin-embedded (FFPE) tissue specimens. The antibodies and reagents used in the experiments are listed as follows: ACTH (Abcam, ab199007), POMC (ProteinTech, 66358-1-Ig), TH (Abcam, ab112), CRH (ProteinTech, 10944-1-AP), CgA (ProteinTech, 60135-1-Ig), and Human Galanin Antibody (R&D, MAB5854).

## Acknowledgements

This work was supported by CAMS Innovation Funds for Medical Sciences (CIFMS), which were 2017-I2M-1-001, 2021-I2M-1-051 and 2021-I2M-1-001.

## Additional information

### Funding

| Funder | Grant reference number | Author |
| --- | --- | --- |
| Chinese Academy of Medical Sciences | 2017-I2M-1-001 | Hanzhong Li |
| Chinese Academy of Medical Sciences | 2021-I2M-1-051 | Taijiao Jiang |
| Chinese Academy of Medical Sciences | 2021-I2M-1-001 | Taijiao Jiang |

The funders had no role in study design, data collection and interpretation, or the decision to submit the work for publication.

### Author contributions

Xuebin Zhang, Data curation, Formal analysis, Investigation, Methodology, Resources, Software, Writing – original draft, Writing – review and editing; Penghu Lian, Formal analysis, Investigation, Methodology, Project administration, Resources, Software, Validation, Visualization, Writing – original draft, Writing – review and editing; Mingming Su, Data curation, Formal analysis, Investigation, Methodology, Resources, Software, Validation, Visualization, Writing – original draft, Writing – review and editing; Zhigang Ji, Data curation, Investigation, Methodology, Visualization, Writing – review and editing; Jianhua Deng, Data curation, Investigation, Methodology, Writing – review and editing; Guoyang Zheng, Wenda Wang, Data curation, Investigation, Writing – review and editing; Xinyu Ren, Data curation, Visualization; Taijiao Jiang, Hanzhong Li, Conceptualization, Funding acquisition, Project administration, Supervision, Writing – review and editing; Peng Zhang, Investigation, Methodology, Supervision, Validation, Writing – original draft, Writing – review and editing

### Author ORCIDs

Mingming Su http://orcid.org/0000-0002-1393-0800
Peng Zhang http://orcid.org/0000-0002-6218-1885

### Ethics

Ethics approval and consent to participateSpecimen collection was obtained after appropriate research consents (and assents when applicable) and was approved (protocol number: S-K431) by the Institutional Review Board, Peking Union Medical College Hospital. All information obtained was protected and de-identified.

### Decision letter and Author response

Decision letter https://doi.org/10.7554/eLife.68436.sa1
Author response https://doi.org/10.7554/eLife.68436.sa2

## Additional files

### Supplementary files

• Supplementary file 1. Raw data QC for single-cell RNA sequencing and whole-exome sequencing. xlsx.

• Supplementary file 2. Number (n) of cells in each cellular cluster in different samples and top 10 gene markers.xlsx.

• Supplementary file 3. Number (n) of cells in each endothelial cell cluster in different samples and top 10 gene markers.xlsx.

• Supplementary file 4. Germline and somatic mutations of tumor and normal samples from the rare case with ectopic ACTH&CRH-secreting pheochromocytoma by whole-exome sequencing.xlsx.

• Transparent reporting form

## Data availability

The raw data of scRNA-seq sequencing reads generated in this study were deposited in The National Genomics Data Center (NGDC, https://bigd.big.ac.cn/) under the accession number: PRJCA003766.

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

# Appendix 1

## Clinical samples description

Case 1: A 39-year-old lady underwent laparoscopic left adrenal tumor resection in July 2012 at a local hospital. She had a 2-year history of headache, generalized swelling, and palpitations. She was noted to have hypertensive (BP 240/120 mmHg) and typical Cushingoid characteristics, including asthenia, supraclavicular fat deposits, bruises, purple striae, proximal myopathy, and hyperpigmentation. Histopathology confirmed an adrenomedullary chromaffin tumor. During tumor immunostaining, the tumor stained positively for ACTH. After the adrenal surgery, her Cushingoid characteristics, hypokalemia, and hypertension were all relieved.

However, the patient experienced recurrence of symptoms and signs in January 2019 and was admitted to our hospital. It was found that urine and plasma metanephrine were significantly elevated, and plasma ACTH was also high. Enhanced CT scanning of the abdomen revealed bilateral adrenocortical hyperplasia and multiple masses in the left adrenal and around the left kidney. The largest mass lesion was 2.3×1.6 cm$^2$, which invaded upper pole of left kidney. But the I123-MIBG scintigraphy was negative. We performed a surgery to remove left adrenal, kidney, and masses. After the surgery, the patient's clinical features and symptoms were improved, and the excessive hypercortisolemia and catecholamine eventually returned to normal. IHC revealed positive staining for chromogranin A, ACTH, and CRH, confirming the diagnosis of pheochromocytoma secreting both ACTH and CRH.

Case 2: A 42-year-old male with a 3-year history of headache and palpitations, and a 6-month history of hypertension was admitted to our hospital. Laboratory tests showed that the plasma and urine catecholamines and their metabolites were elevated, and cortisol and ACTH were at the normal level. Enhanced CT showed a 67×70 mm$^2$ left adrenal tumor, and I123-MIBG scintigraphy exhibited positive. We performed a surgery to remove the left adrenal gland. After the surgery, the patient's clinical features and symptoms were relieved. IHC confirmed the diagnosis of pheochromocytoma.

Case 3: A 50-year-old female came to our hospital with hypertension, hyperkalemia, and Cushingoid symptoms (moon face and central obesity). Enhanced CT scanning revealed a 19×36 mm$^2$ irregular mass in left adrenal gland. The laboratory tests showed ACTH-independent hypercortisolemia. The left adrenal gland was removed, and Cushing's syndrome was relieved. Resected specimen revealed two tumors in the left adrenal gland, and IHC confirmed the diagnosis of adrenal adenoma.

**Appendix 1—table 1.** Summary of laboratory test for three cases.

| Laboratory test | Case 1 | Case 2 | Case 3 | Reference range |
|---|---|---|---|---|
| ACTH | 519.0 | 24.0 | <5 | 0–46.0 pg/ml |
| 24 hr urine-free cortisol | 2024.4 | | 332.4 | 12.3–103.5 µg/24 hr |
| Catecholamines | | | | |
| Plasma metanephrines | | | | |
| Normetanephrine | 3.28 | 10.81 | 0.4 | <0.9 nmol/L |
| Metanephrine | 3.44 | 11.55 | 0.2 | <0.5 nmol/L |
| 24 hr urine | | | | |
| Epinephrine | 397.63 | 56.23 | 1.92 | 1.74–6.42 µg/24 hr |
| Norepinephrine | 475.43 | 82.29 | 26.17 | 16.69–40.65 µg/24 hr |
| Dopamine | 432.21 | 301.71 | 240.5 | 120.93–330.5 µg/24 hr |

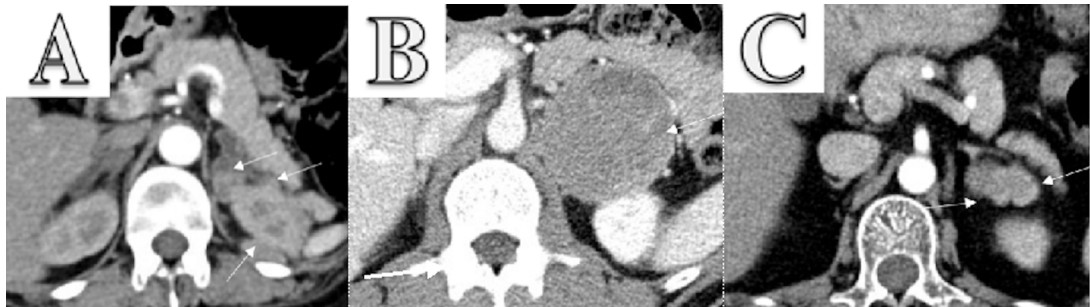

**Appendix 1—figure 1.** Enhanced CT scanning image for three cases. (**A**) Enhanced CT scanning for Case 1 with pheochromocytoma secreting both ACTH and CRH. The abdomen revealed bilateral adrenocortical hyperplasia and multiple masses in the left adrenal and around the left kidney. The largest mass lesion was 2.3×1.6 cm$^2$, which invaded upper pole of left kidney. (**B**) Enhanced CT scanning for Case 2 with pheochromocytoma. It showed a 67×70 mm$^2$ left adrenal tumor. (**C**) Enhanced CT scanning for Case 3 with two adrenocortical adenomas. It revealed a 19×36 mm$^2$ irregular mass in left adrenal gland.

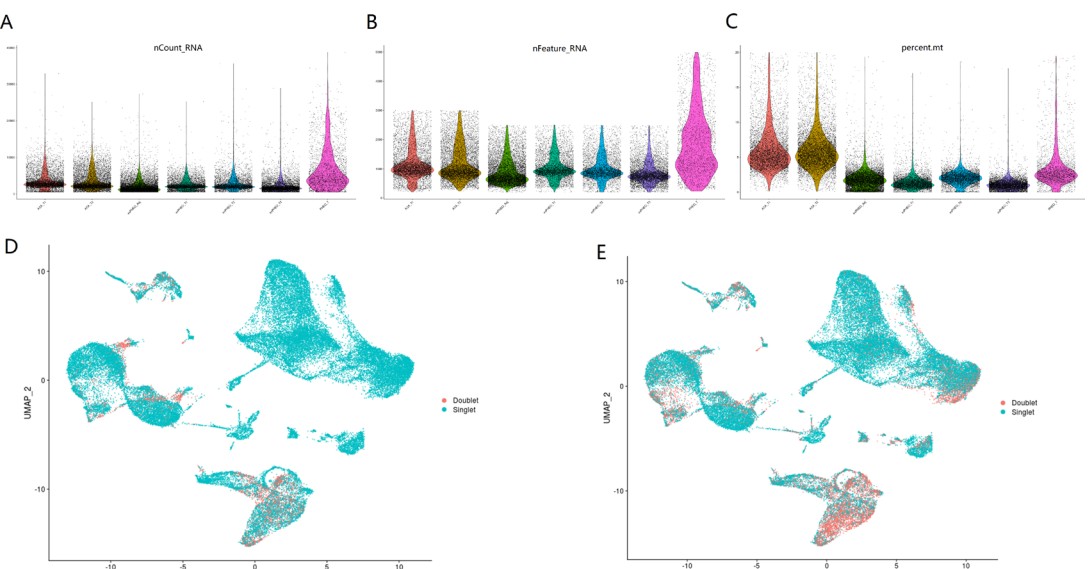

**Appendix 1—figure 2.** Quality control plots and doublet detection for this scRNA-seq study. Violin plots showing number of total RNAs (**A**), number of genes (**B**), and percentage of mitochondrial (mito) genes (**C**) for cells in seven samples. Doublets were predicted by DoubletFinder (**D**) and DoubletDecon (**E**).

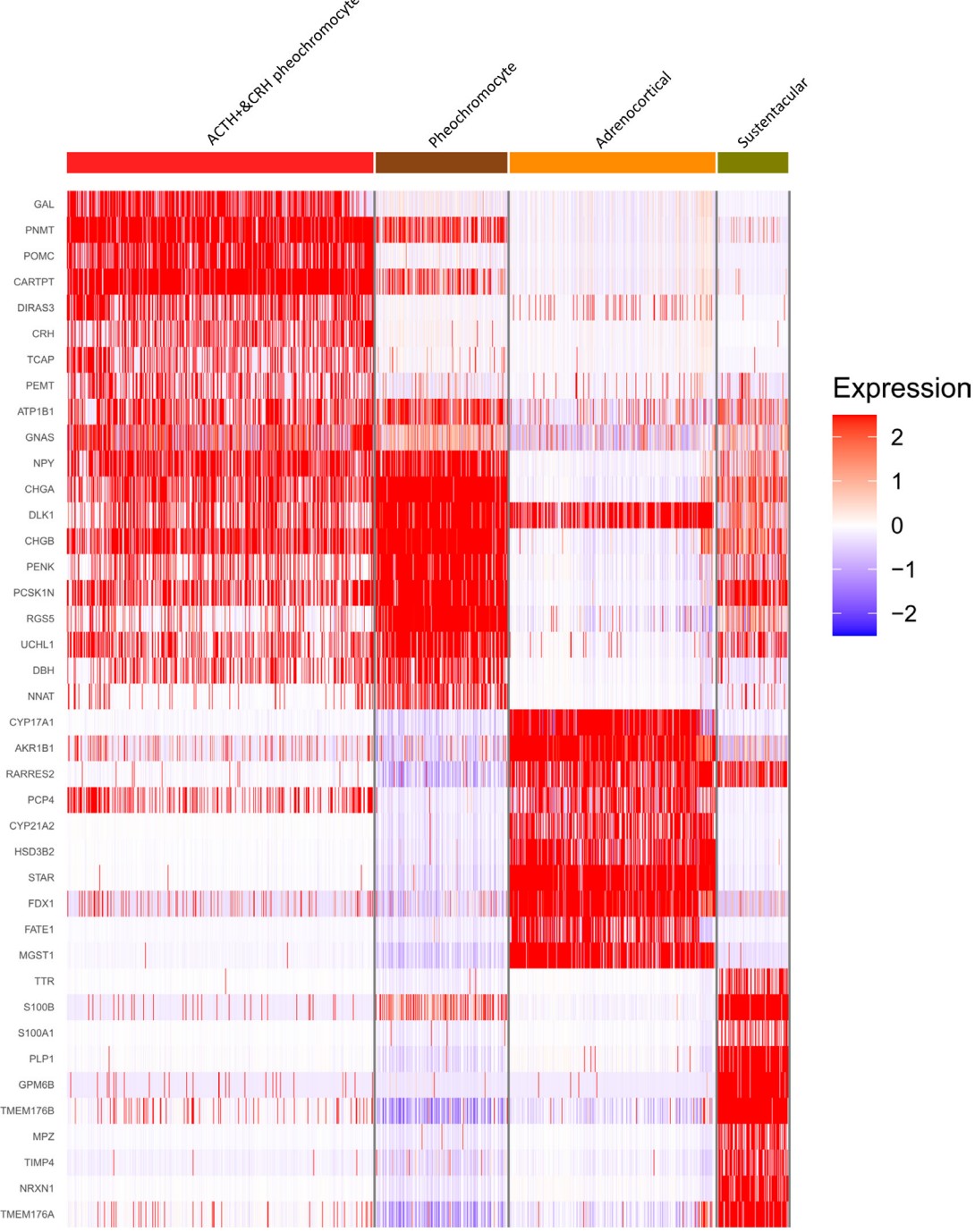

**Appendix 1—figure 3.** Four adrenal cell types and their highly expressed genes through single-cell transcriptomic analysis. Heatmap shows the scaled expression patterns of top 10 marker genes in each cell type. The color keys from white to red indicate relative expression levels from low to high.

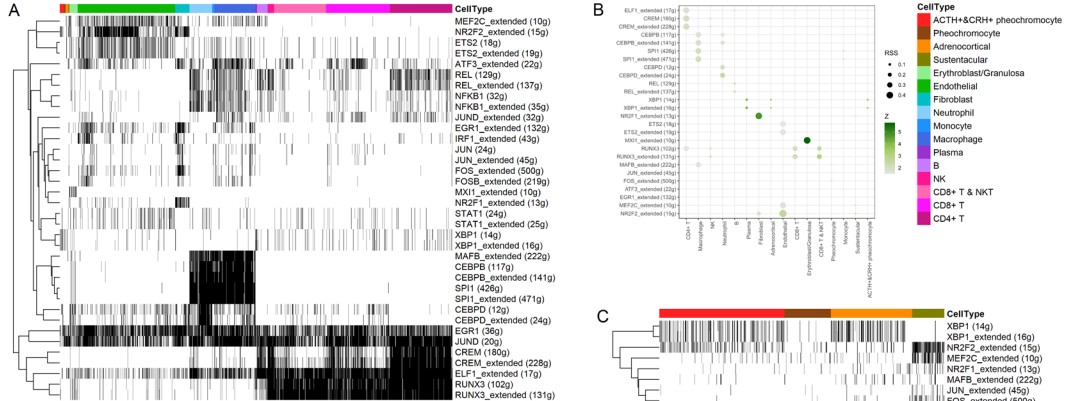

**Appendix 1—figure 4.** Transcription factors detection using SCENIC pipeline. (**A**) Binarized heatmap showing the AUC score (area under the recovery curve, scoring the activity of regulons) of the identified regulons plotted for each cell. (**B**) For each cellular cluster, dot plot showed regulons were selected based on regulon specificity score (RSS). (**C**) Binarized heatmap showed specific regulons for four types of adrenal cells and found XBP1 as the top regulons for ACTH+&CRH + pheochromocyte.

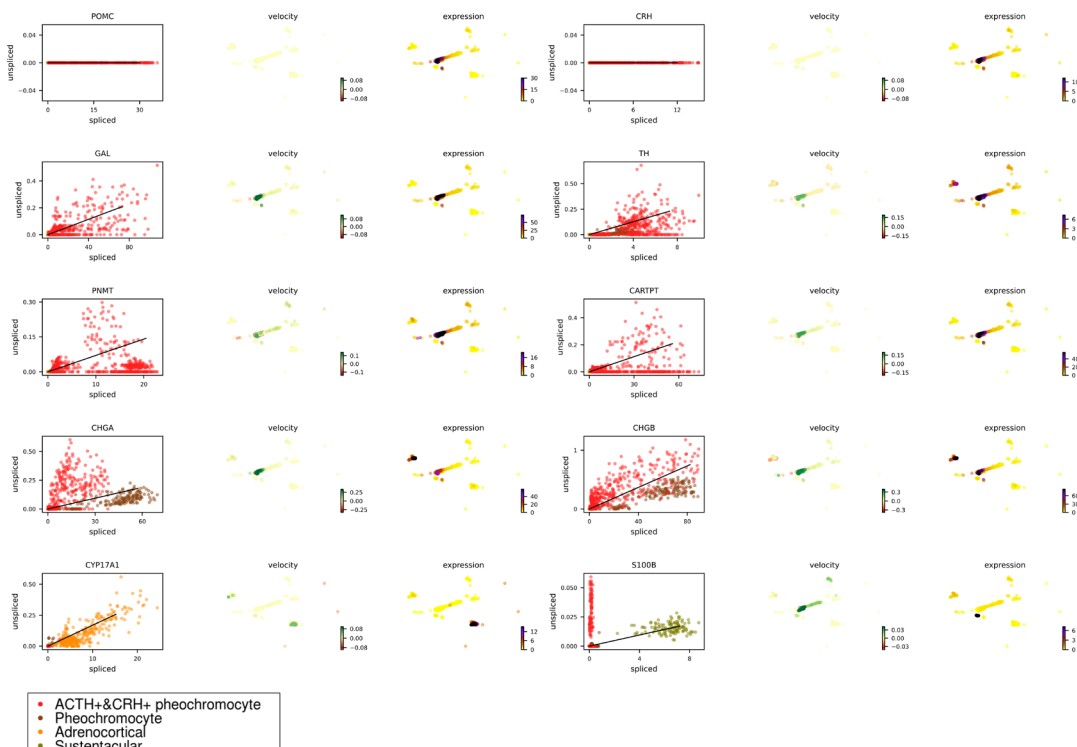

**Appendix 1—figure 5.** The spliced versus unspliced phase for marker genes in four types of adrenal cells. Transcripts were marked as either spliced or unspliced based on the presence or absence of intronic regions in the transcript. For each gene, the scatter plot shows spliced and unspliced ratios in a single cell. Each point corresponds to a cell, colored by different adrenal cellular types. A simple model of RNA dynamics is fit to the data. The estimation of RNA velocities and expression for each cell, visualized in UMAP plot. UMAP, Uniform Manifold Approximation and Projection.

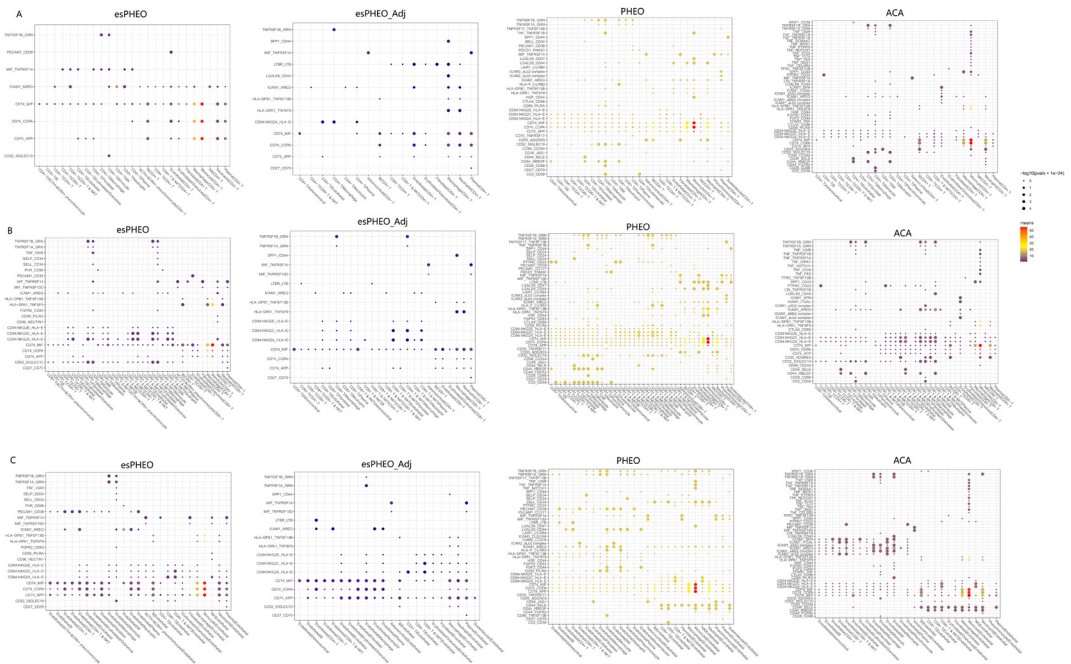

**Appendix 1—figure 6.** Ligand-receptor interaction analysis for CD4+ T cells, CD8+ T cells, and endothelial cells in different tumor microenvironments. Overview of ligand-receptor interactions between the CD4+ T cells (**A**), CD8+ T cells (**B**), endothelial (**C**), and the other cell types in the different tumor microenvironments. p-values are represented by the size of each circle. The color gradient indicates the level of interaction.

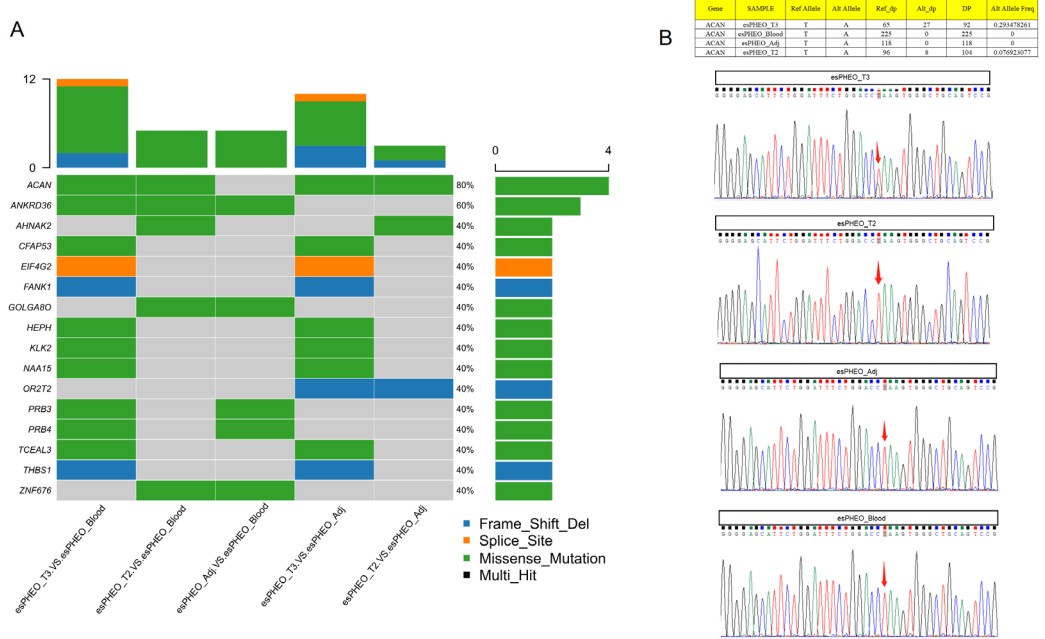

**Appendix 1—figure 7.** Whole-exome sequencing identified one shared somatic variant of *ACAN* comparing variants in tumor samples to controls and Sanger sequencing only confirmed the presence in esPHEO_T3 but not observed in esPHEO_T2. (**A**) Distribution of somatic mutations for the rare case
*Appendix 1—figure 7 continued on next page*

*Appendix 1—figure 7 continued*

with ectopic ACTH&CRH-secreting pheochromocytoma. OncoPrint plots were generated using the R package Maftools for somatic mutations from five tissue comparisons: esPHEO_T3 versus esPHEO_Blood, esPHEO_T2 versus esPHEO_Blood, esPHEO_Adj versus esPHEO_Blood, esPHEO_T2 versus esPHEO_Adj, and esPHEO_T3 versus esPHEO_Adj. (**B**) Sanger sequencing to validate the somatic mutation of *ACAN_c.5951T > A:p.L1984Q* located on chr15:88858536. It confirmed the presence in esPHEO_T3 but not validated in esPHEO_T2. The variant was not observed in controls (esPHEO_Adj and esPHEO_Blood).

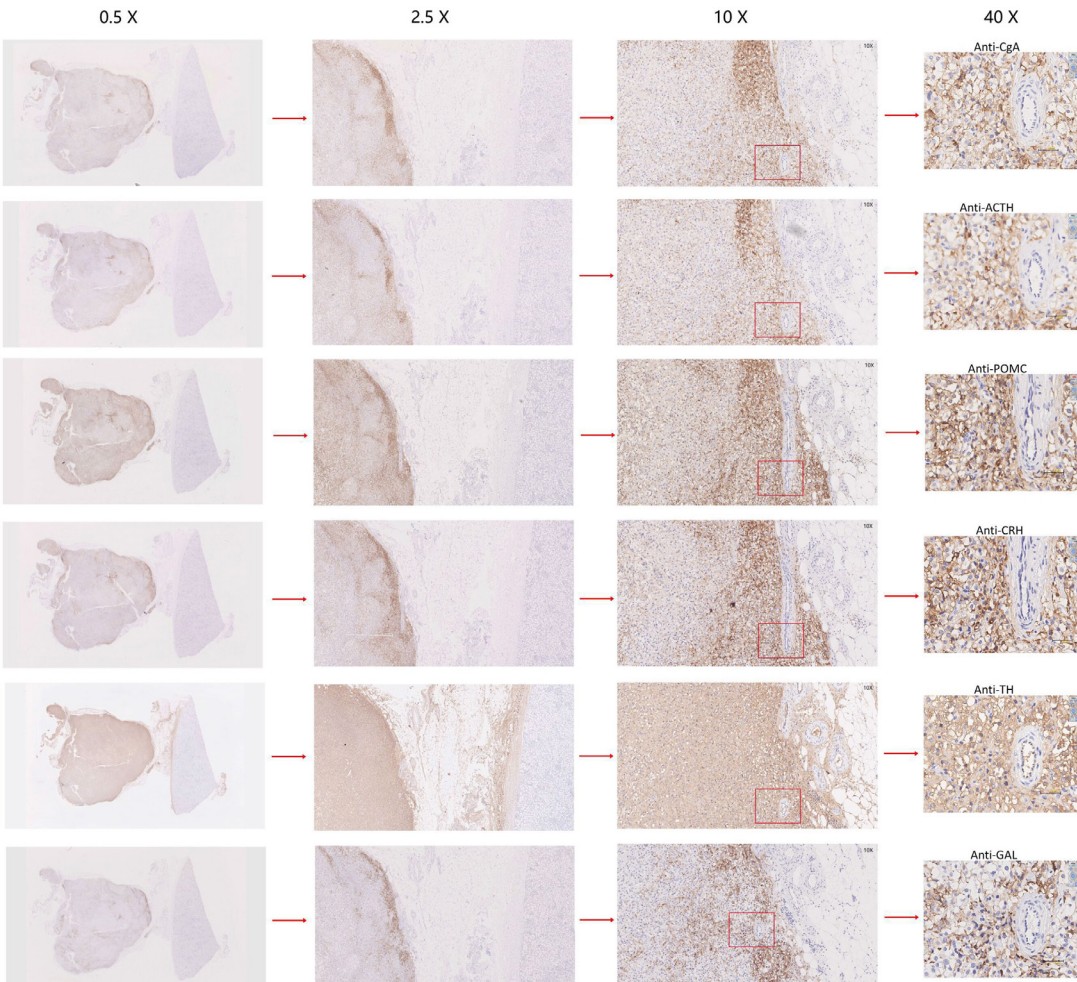

**Appendix 1—figure 8.** Immunohistochemistry of CgA, ACTH, POMC, CRH, TH, or GAL on serial biopsies from tumor specimen infiltrating tissues located in the kidney (esPHEO_T3). We observed positive staining signal at tumor left in each slice, while the adjacent kidney was un-stained could be negative controls. The magnification is 0.5×, 2.5×, 10×, and 40× from left to right. Red rectangular indicates the magnified area of the location, as shown in *Figure 3D*.

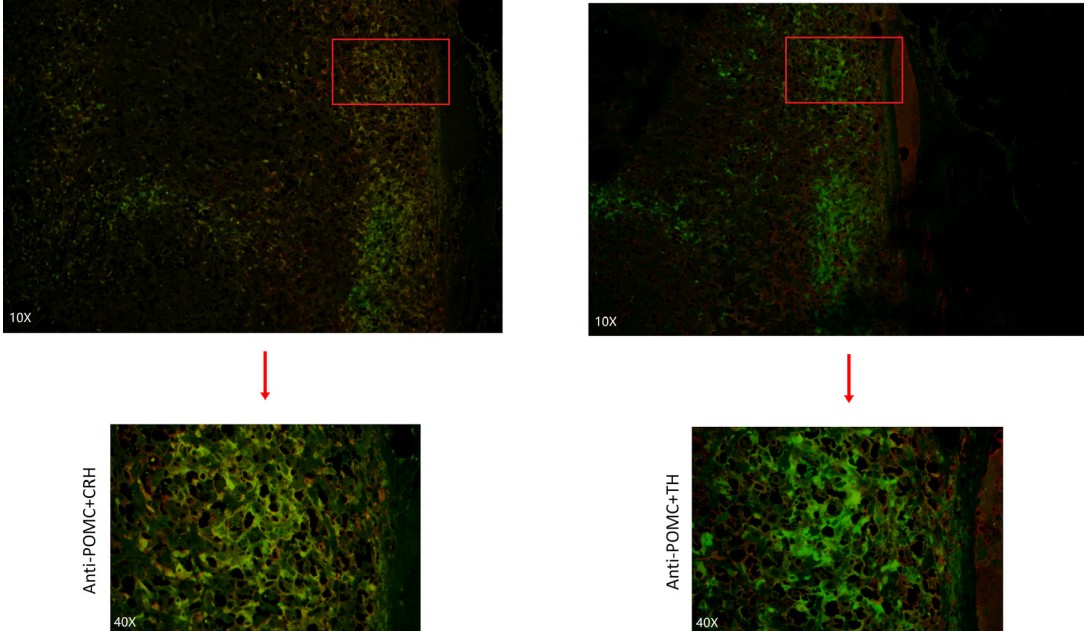

**Appendix 1—figure 9.** Immunofluorescence co-staining for POMC&CRH and POMC&TH on two serial biopsies from tumor specimen esPHEO_T3. The magnification is 10× (top) and 40× (bottom). Red rectangular indicates the magnified area of the location, as shown in *Figure 3E*.

