## [Editor Report]

The study described an extremely rare type of adrenal pheochromocytoma that secretes both ACTH and CRH, in addition to catecholamines. Single-cell RNA sequencing of the tumor and other tumors revealed a group of cells that are responsible for the hormone secretion. We believe that this work will provide an interesting example of functional endocrine tumors and how they are formed.

---

## [Decision Letter]

**Decision letter after peer review:**

Thank you for submitting your work entitled "Single-cell transcriptome analysis identifies a unique tumor cell type producing multiple hormones in ectopic ACTH and CRH secreting pheochromocytoma" for further consideration by *eLife*. Your article has been reviewed by 3 peer reviewers, one of whom is a member of our Board of Reviewing Editors, and the evaluation has been overseen by Mone Zaidi as the Senior Editor.

*Reviewer #1:*

The authors identified an extremely rare case of ATCH-dependent Cushing syndrome due to ACTH&CRH secreting pheochromocytoma. They retrieved sugically resected samples from the tumor and subjected them to scRNA-seq, which led them to identify a group of cells that are double-positive for ACTH&CRH. They then performed a series of expriments to confirm that the cells are indeed present in the tissue, and attempted to identify genes that may lie upstream of the process.

Perhaps the most important point of the study is the identification of the double-positive (DP) cells from the patient. However, evidence supporting this observation is relatively scarce other than showing a cell cluster that express POMC, CRH etc (as displayed in Figure 3A, C). Gene expression pattern shown in Figure 3C supports that the DP cells share molecular characteristics with those of pheochromocytes. But in the t-SNE plot, these cells are located far from pheochromocytes in PHEO_T. Rather, the DP cell cluster seems to be branched out from immune cells. If I didn't read the t-SNP plot wrong, I wonder why the identity of DP cells is closer to the immune cells. Also, it needs to be clarified if the DP cells could be doublets? The authors did not show basic statistics and QA/QC data of the scRNA-seq experiment (as supplementary data for example). They should show that the DP cells are not technical doublet cells.

Another critical question would be what is the genetic driver that induces expression of both hormones in the DP cells? They propose GAL, but the evidence supporting its direct role is not strong and remains speculative.

Comments for the authors:

Overall, this study requires more carefully designed expriments and interpretation. Otherwise, it remains as a descriptive study with vague conclusions, leaving the uniqueness of the sample being the only strength of the study.

1. Colors in Figure 3A are confusing.

2. Figure 5 does not add much to the molecular mechanism. Rather it merely describes physiological consequences by the presence of DP cells. Please consider strengthen or remove it.

3. Isn't Figure 7B a duplication of Figure 3B?

4. IHC data in Figure 3E, F lack negative controls. And the readers need additional markers to be guided of its anatomical location.

5. Figure 4 compared DEGs between DP cells and other tumor cells. Since the cell groups that were being compared are too different, observing such dramatic differences is not unexpected and hard to coin physiological relevance. Wouldn't it be more meaningful to compare them to pheochromocytes?

6. The pseudotime analysis in Figure 6 does not answer the question of how the DP cells originated. It should be performed in a such way to suggest genes that marks critical points during the pseudotime branching or proceeding.

*Reviewer #2:*

In this manuscript Zhang et al. generated single cell RNA sequencing data for the adrenal gland tumors including extremely rare type of tumor, ACTH & CRH-secreting pheochromocytoma. Unbiased clustering analysis discovered a unique tumor cell type that expresses multiple hormones unlike normal adrenal gland cells and other tumor cell types that produce a single hormone. By comparing with other type of tumor cells, they identified specific marker genes of the novel tumor cell type. They also revealed the distinct immune and endothelial cell populations in the microenvironment of different tumor samples.

Although the gene expression profiles of novel cell type can be utilized to reveal the molecular mechanism of this rare tumor associated with Cushing's syndrome, the data was generated from only a single patient and have not validated in other samples. In addition, the results only provide the list of genes that were specifically expressed in the novel tumor cell type and their potentially related biological pathways, but not detail molecular and cellular characters of the cells. The single cell gene expression profiling data are definitely useful for the researches.

Comments for the authors:

I have several concerns and suggestions, which if addressed would improve the manuscript.

1. The major finding of this manuscript is the presence of multi-functional tumor cell type which produce multiple hormones such as POMC, the precursor of ACTH and CRH. But, this finding was only derived from a single sample and experimentally validated using the same tissue. I understand the sample is very rare, but could the authors validate the result in different tumor samples at least using IHC or IF? If sample is not available, the limitation of the study should be mentioned.

2. Please consider providing full list of marker genes that were used for cell type annotation.

3. Figure 3C does not seem to support the statement "We demonstrated that GAL was expressed in the ACTH^+^&CRH^+^ pheochromocyte and 'regulated the secretion of ACTH'".

4. The authors identified a unique and important multi-functional cell type but current analyses (differentially expressed genes identification and gene ontology analysis) seem insufficient to characterize molecular feature of ACTH^+^&CRH^+^ pheochromocyte. The authors could perform additional comprehensive analysis such as SCENIC analysis in order to identify the master transcription regulator of the cell type.

5. The pseudo-time analysis indicated that sustentacular cells transform to ACTH^+^&CRH^+^ pehochromocytes and then to pheochromocyte. The authors utilized Monocle3 in which user has to define the starting points. The authors can validate the result using RNA velocity analysis which also predicts cell transition without the need of prior knowledge about starting point cell type.

6. Given the diverse immune and endothelial cell type in the tumor microenvironment, it would be interesting to perform the cell-cell interaction analysis using the programs such as CellPhoneDB to see if they have distinct regulatory role in different tumor microenvironment.

7. How did the authors define the four subclusters of endothelial cells? Please consider providing list of marker genes.

8. In the method part, how did the authors determine different criteria for the maximum number of genes (no more than 5000, 3000, and 2500 genes for PHEO, ACA, and esPHEO samples, respectively)?

*Reviewer #3:*

Zhang et al. perform single cell RNA sequencing (scRNA-Seq) of one rare ACTH^+^CRH-secreting phenochromocytoma (3 anatomically distinct sites from the tumor and one peritumoral site), one typical pheochromocytoma, and two typical adrenocortical adenomas.

Their main findings are as follows: (1) They identify a unique cell type, which they term ACTH^+^CRH^+^ pheochromocyte, which appears to be the tumor cell present in the rare ACTH^+^CRH^+^ tumor (2) Marker gene analysis reveals that while known adrenal chromaffin markers (CHGA, PNMT) are present in both pheochromocytes and ACTH^+^CRH^+^ pheochromocyte, the latter has some unique markers such as GAL and POMC. They validate the marker genes with IHC. (3) Profiling of the non-tumor populations reveals distinct immune microenvironment profile and endothelial cell profile to the rare tumor compared with classical pheochromocytoma and adrenalocortical adenoma.

The main strength of this manuscript is that it involves single-cell profiling of an exceptionally rare tumor type and a distinction from the more common adrenal tumors (pheochromocytoma and adrenocortical adenoma). The broader implication of the authors' findings is with respect to Dale's principle, which states that a given neuron releases only one type of neurotransmitter. However, in the case of this tumor, single cell analysis clearly shows that ACTH, CRH, and chatacholemines are being released from the same cell. This is quite interesting and significant. The data will also potentially be valuable to others in the field for analysis in future studies.

There remain some unanswered questions – namely:

(1) What is the cell in normal physiology that gives rise to this ACTH^+^CRH^+^ pheochromocytoma?

(2) Do conventional phenochromocytomas differ from the ACTH^+^CRH^+^ pheochromocytoma in terms of the cell of origin that is transformed, or in the spectrum of genetic alterations that result in transformation?

Comments for the authors:

Overall, I think this study is of broad interest given the rarity of this tumor type. My comments to the authors to improve the manuscript are as follows:

1. Given how rare the ACTH^+^CRH^+^ pheochromocytoma is, I think the study would be substantially strengthened if the authors could perform DNA sequencing (WGS or WES) and describe how, if at all, the genomic landscape differs from conventional pheochromocytoma.

2. Can the authors comment on whether the hypothesis is whether the ACTH^+^CRH^+^ pheochromocytoma originates from a rare progenitor cell that is distinct from the chromaffin cell giving rise to pheochromocytoma? If so, can the authors stain a panel of normal adrenal glands with some of their marker genes to try and identify this cell in normal tissues?

3. While the tumor type is interesting for its rarity, the analysis performed is quite standard and comes across as a bit superficial in parts. Although it is understandable that the authors have only one ACTH^+^CRH^+^ sample I think they can do more with the data and this would significantly strengthen the manuscript. For example, it would be interesting if the authors can point to specific master regulatory factors that drive the distinct programs in pheochromocytes vs. ACTH^+^CRH^+^ pheochromocytes. The immune microenvironment analysis, while inherently descriptive, is also somewhat superficial.

[Editors' note: further revisions were suggested prior to acceptance, as described below.]

Thank you for submitting your revised article "Single-cell transcriptome analysis identifies a unique tumor cell type producing multiple hormones in ectopic ACTH and CRH secreting pheochromocytoma" for consideration by *eLife*. Your article has been reviewed by 3 peer reviewers, including Murim Choi as the Reviewing Editor and Reviewer #1, and the evaluation has been overseen by Mone Zaidi as the Senior Editor.

Essential revisions:

Although the reviewers thought that many issues were addressed, they still concerned on the superficial analysis results. Nonetheless, they agreed that the manuscript contains a common interest for publication in *eLife* as the tumor is an extremely rare case. Please address reviewers' concerns below.

*Reviewer #1:*

Although the authors could not address all the questions, especially regarding the origin of DP cells and genetic driver for DP cells, it appears reasonable that they are hard to address as the tumor sample was extremely rare.

*Reviewer #2:*

Although the authors have satisfactorily addressed most of my points, there are remaining concerns about RNA velocity data.

Please cite any reference for the statement "For the high proportions of unspliced/spliced transcripts, stem-like characteristics of sustentacular cells were supported." Can global ratio of unspliced/spliced transcripts support stem-like characteristics?

Please elaborate Figure 5 C-F. Currently, they don't seem to add any information.

*Reviewer #3:*

In the revised manuscript Zhang et al. have included additional data and analyses including more exhaustive QC, RNA velocity analysis, regulome analysis, and have performed WES of the ACTH/CRH-secreting pheochromocytoma. They have generally addressed my technical concerns from the prior review. I maintain that the analysis remains somewhat superficial and descriptive in parts and this may be somewhat of a missed opportunity to more deeply explore the underlying biology of this unique case, understanding the caveats of its rarity. Nonetheless, I think a description of this tumor at single-cell resolution and availability of the dataset is of value to the scientific community.

However, I would like to see a more careful analysis of the WES data prior to publication. I do not see any basic metrics (mutation rate etc.), description of pathogenicity filtering/annotation, or copy number analysis. The mutations shown are primarily missense and I do not really see any obvious driver genes – how many of these are putative driver vs. passenger mutations? ACAN is mentioned, but what is its significance, if any? The somatic landscape should be discussed in comparison to typical phenochromocytomas and adrenocortical carcinomas, which have been more extensively sequenced. If there is no obvious genetic driver of this ACTH/CRH-secreting phenochromocytoma, that should be stated. If the claim is that ACAN alterations are somehow related to this tumor type, that needs to be substantiated. Or if the implication is that ACAN is a passenger alteration, that needs to be stated explicitly also.

---

## [Author Response]

Reviewer #1:The authors identified an extremely rare case of ATCH-dependent Cushing syndrome due to ACTH&CRH secreting pheochromocytoma. They retrieved surgically resected samples from the tumor and subjected them to scRNA-seq, which led them to identify a group of cells that are double-positive for ACTH&CRH. They then performed a series of experiments to confirm that the cells are indeed present in the tissue, and attempted to identify genes that may lie upstream of the process.

We thank the reviewer for carefully reviewing the manuscript. We updated graphs, added supplementary files of raw data QC and cell cluster statistics, and performed RNA velocity analysis, scenic analysis for the single cell RNA sequencing experiments to response the reviewer’s critiques and strengthen the manuscript. In addition, to investigate the genetic driver for Case 1, we supplemented whole-exome sequencing experiments for all rest specimens, that is, tumors (esPHEO_T2, esPHEO_T3) and controls (esPHEO_Adj, esPHEO_Blood) from the rare case with ectopic ACTH&CRH-secreting pheochromocytoma.

Perhaps the most important point of the study is the identification of the double-positive (DP) cells from the patient. However, evidence supporting this observation is relatively scarce other than showing a cell cluster that express POMC, CRH etc (as displayed in Figure 3A, C). Gene expression pattern shown in Figure 3C supports that the DP cells share molecular characteristics with those of pheochromocytes. But in the t-SNE plot, these cells are located far from pheochromocytes in PHEO_T. Rather, the DP cell cluster seems to be branched out from immune cells. If I didn't read the t-SNP plot wrong, I wonder why the identity of DP cells is closer to the immune cells. Also, it needs to be clarified if the DP cells could be doublets? The authors did not show basic statistics and QA/QC data of the scRNA-seq experiment (as supplementary data for example). They should show that the DP cells are not technical doublet cells.

We thank the reviewer for raising the concerns and providing these helpful suggestions. First, we updated the colors mapped to 16 cellular clusters in Figure 2A and Figure 3A to enhance the color difference between doublet-positive (DP) cells and immune cells. Then, the new analysis based on RNA velocity was performed in the revision, and the results showed that DP cluster was isolated and not branched out from other cell types (including immune cells) from velocity streamlines (Figure 5F). In addition, we added the raw data QC and doublet prediction results of the scRNA-seq experiment as shown in Appendix 1—figure 2 and Supplementary File 1. From the doublets predicted by DoubletFinder and DoubletDecon, it is clarified that almost noDP cells were defined as doublets. Cellular cluster statistics were shown in Supplementary File 2, which presented cell counts for each cellular cluster in different samples and top10 gene markers.

Another critical question would be what is the genetic driver that induces expression of both hormones in the DP cells? They propose GAL, but the evidence supporting its direct role is not strong and remains speculative.

We thank the reviewer for raising these important concerns, and we agree with the reviewer that the presentation about the genetic driver in the previous version of the manuscript is not sufficient enough. We changed the conclusion statement "We demonstrated that GAL was expressed in the ACTH^+^&CRH^+^ pheochromocyte and regulated the secretion of ACTH" to "We demonstrated that GAL was expressed in the ACTH^+^&CRH^+^ pheochromocyte and might participate in the regulation of ACTH secretion". (Page 7 line 175-182)

We provided more description and additional analysis about putative genetic driver in the DP cells, as follows:

First, we found GAL co-expressed with POMC and CRH, could be a candidate marker to detect the rare ectopic ACTH^+^&CRH^+^ secreting pheochromocytes. It might be involved in the regulation of the hypothalamic-pituitary-adrenal axis. (Page 7 line 175-182, Figure 3, Figure 4).

Second, we also found an additional weak signal of transcription regulons for the DP cells (Page 6 line 153-157, Appendix 1—figure 4). It showed *XPBP1* as the specific regulons for ACTH^+^&CRH^+^ pheochromocyte and adrenocortical cell type.

Third, to investigate the genetic driver, we supplemented whole-exome sequencing experiments for tumors (esPHEO_T2, esPHEO_T3) and controls (esPHEO_Adj, esPHEO_Blood) from the rare case with ectopic ACTH&CRH-secreting pheochromocytoma. We identified 1 shared somatic variant of *ACAN* (*c.5951T>A:p.L1984Q*) comparing variants in tumor samples to controls but Sanger sequencing only confirmed the presence in esPHEO_T3 which was not observed in esPHEO_T2 (Page 13 line 352-358, Appendix 1—figure 7).

Comments for the authors:Overall, this study requires more carefully designed experiments and interpretation. Otherwise, it remains as a descriptive study with vague conclusions, leaving the uniqueness of the sample being the only strength of the study.

We thank the reviewer for carefully reviewing and helpful suggestions. We updated graphs and tables, implemented supplementary analysis for the single-cell RNA sequencing data. Because this case is particularly rare, fresh tissue samples are lacking, currently, frozen tissue samples cannot be assayed by flow cytometry. For all rest of the samples, we can only supplement the whole-exome sequencing experiments for tumors (esPHEO_T2, esPHEO_T3) and controls (esPHEO_Adj, esPHEO_Blood) from the rare case with ectopic ACTH&CRH-secreting pheochromocytoma to make our results more comprehensive. Lastly, on one hand, we are very concerned about similar suspicious cases in the clinic. On the other hand, we are going for the following research for further downstream experiments to validate the molecular mechanism for secreting multiple hormones.

1. Colors in Figure 3A are confusing.

We have updated the colors mapped to 16 cellular clusters in Figure 2 and Figure 3 to enhance the color difference between doublet-positive (DP) cells and immune cells.

2. Figure 5 does not add much to the molecular mechanism. Rather it merely describes physiological consequences by the presence of DP cells. Please consider strengthen or remove it.

Due to the previous Figure 5 mainly describe the physiological consequences by the presence of DP cells as the reviewer commented. We have moved it to Figure 4D, because the differential expressed genes between DP cells and other adrenal cell types were shown in Figure 4A and Figure 4C. Combining these figures into a group could complement each other and clarify the secreting functions of the DP cells.

3. Isn't Figure 7B a duplication of Figure 3B?

Figure 3B presents the frequency distribution of all cell types among different samples, while in Figure 7B we specifically focused on the immune microenvironments and showed statistics of immune cell types. To some extent, they are repetitive since both describe the percentage of immune cells. But the denominators are different for percentage calculation, that is, one is the total number of cells in Figure 3B, the other is the total number of immune cells in Figure 7B.

4. IHC data in Figure 3E, F lack negative controls. And the readers need additional markers to be guided of its anatomical location.

We supplemented IHC figures of CgA, ACTH, POMC, CRH, TH or GAL with magnification (0.5x, 2.5x, 10x, 40x) from tumor specimen infiltrating tissues located in the kidney (esPHEO_T3) in Appendix 1—figure 8. We observed positive staining signal at tumor left in each slice, while the adjacent kidney was un-stained could be negative controls. Red rectangular indicates the magnified area of the location as shown in Figure 3D. The. We supplemented the immunofluorescence (IF) co-staining figures with magnification (10x, 40x) for POMC&CRH and POMC&TH from tumor specimen esPHEO_T3 in Appendix 1—figure 9, where red rectangular indicates the magnified area of the location in Figure 3E.

5. Figure 4 compared DEGs between DP cells and other tumor cells. Since the cell groups that were being compared are too different, observing such dramatic differences is not unexpected and hard to coin physiological relevance. Wouldn't it be more meaningful to compare them to pheochromocytes?

We analyzed the differentially expressed genes (DEGs) between ACTH^+^&CRH^+^ pheochromocyte and the other two subtypes of adrenal tumor cells (pheochromocyte and adrenocortical cells) (Page 9 line 241-245). Such dramatic differences were observed because we set the statistically significant differences as a cut-off p-value < 0.05 and a fold change ≥ 1.5 ( which means a log2 fold change |logFC| ≥ 0.585 ) (Figure 4A). It could more strict such as a cut-off p-value <0.01 and a fold change ≥ 2 ( which means a log2 fold change |logFC| ≥ 1 ). But the top significantly differentially expressed genes were POMC, CRH, GAL etc, as marked in Figure 4A. There is a relatively larger difference in gene expression between DP cells and adrenocortical cells than that between DP cells and pheochromocytes (Figure 4C). Since we didn’t identify any pheochromocytes in esPHEO_adj, we could not compare the DP cells to their adjacent pheochromocytes (Supplementary File 2).

Reviewer #2:In this manuscript Zhang et al. generated single cell RNA sequencing data for the adrenal gland tumors including extremely rare type of tumor, ACTH & CRH-secreting pheochromocytoma. Unbiased clustering analysis discovered a unique tumor cell type that expresses multiple hormones unlike normal adrenal gland cells and other tumor cell types that produce a single hormone. By comparing with other type of tumor cells, they identified specific marker genes of the novel tumor cell type. They also revealed the distinct immune and endothelial cell populations in the microenvironment of different tumor samples.Although the gene expression profiles of novel cell type can be utilized to reveal the molecular mechanism of this rare tumor associated with Cushing's syndrome, the data was generated from only a single patient and have not validated in other samples. In addition, the results only provide the list of genes that were specifically expressed in the novel tumor cell type and their potentially related biological pathways, but not detail molecular and cellular characters of the cells. The single cell gene expression profiling data are definitely useful for the researches.

We thank the reviewer for carefully reviewing and raising insightful critiques. In this study, we reported a rare case in which ectopic ACTH&CRH-secreting pheochromocytoma in the left adrenal. To identify the hormones-secreting cells, we sent specimens for single-cell transcriptome sequencing immediately after the resection. Thus, the majority of our analysis focused on the validation of novel tumor cell type and their multiple hormones-secreting functions. For future studies, on one hand, we are very concerned about similar suspicious cases in the clinic. On the other hand, we are going for following research for further downstream experiments to validate the molecular mechanism for secreting multiple hormones.

Comments for the authors:I have several concerns and suggestions, which if addressed would improve the manuscript.1. The major finding of this manuscript is the presence of multi-functional tumor cell type which produce multiple hormones such as POMC, the precursor of ACTH and CRH. But, this finding was only derived from a single sample and experimentally validated using the same tissue. I understand the sample is very rare, but could the authors validate the result in different tumor samples at least using IHC or IF? If sample is not available, the limitation of the study should be mentioned.

For the case of ACTH and CRH secreting pheochromocytoma, we performed the surgical resection of the tumor at left adrenal (esPHEO_T1) and its infiltrating tissues located in the kidney (esPHEO_T3) and masses (esPHEO_T2), and obtained 3 tumor specimens. The peritumor sample (esPHEO_Adj) was collected from the left adrenal tissue under the supervision of a qualified pathologist. At first, we performed immunohistochemistry (IHC) staining with chromogranin A (CgA) and ACTH markers for esPHEO_T1 and adjacent specimen (esPHEO_Adj) (Figure 1B). To validate our discovery from scRNA-seq data we implemented IHC of CgA, ACTH, POMC, CRH or TH (Figure 3D) on serial biopsies from another tumor specimen (esPHEO_T3) and added immunofluorescence co-staining for POMC&CRH and POMC&TH on two serial biopsies from esPHEO_T3 (Figure 3E). The frozen tissue of esPHEO_T1 is unavailable and a few remaining for esPHEO_T2. For all rest of tissue samples, we supplemented with the whole-exome sequencing experiments for tumors (esPHEO_T2, esPHEO_T3) and controls (esPHEO_Adj) from the rare case with ectopic ACTH&CRH-secreting pheochromocytoma.

2. Please consider providing full list of marker genes that were used for cell type annotation.

We add row annotations for top10 marker genes at the heatmap showing different cellular clusters and their highly expressed genes (Figure 2B). Cellular cluster statistics were supplemented in Supplementary File 2, which presented cell counts for each cellular cluster in different samples and top10 gene markers.

3. Figure 3C does not seem to support the statement "We demonstrated that GAL was expressed in the ACTH^+^&CRH^+^ pheochromocyte and 'regulated the secretion of ACTH'".

We changed the conclusion sentence to "We demonstrated that GAL was expressed in the ACTH^+^&CRH^+^ pheochromocyte and might participate in the regulation of ACTH secretion". We’re trying to express that: [We found GAL co-expressed with POMC and CRH, could be a candidate marker to detect the rare ectopic ACTH^+^&CRH^+^ secreting pheochromocytes. As previous research reported, it might be involved in the regulation of the hypothalamic-pituitary-adrenal axis.]

4. The authors identified a unique and important multi-functional cell type but current analyses (differentially expressed genes identification and gene ontology analysis) seem insufficient to characterize molecular feature of ACTH^+^&CRH^+^ pheochromocyte. The authors could perform additional comprehensive analysis such as SCENIC analysis in order to identify the master transcription regulator of the cell type.

We have performed additional analysis (Page 18 line 519-570), including RNA velocity analysis, SCENIC analysis etc. In addition, whole-exome sequencing experiments for tumors (esPHEO_T2, esPHEO_T3) and controls (esPHEO_Adj, esPHEO_Blood) from the rare case with ectopic ACTH&CRH-secreting pheochromocytoma were performed to make our results more comprehensive.

First, based on differentially expressed genes identification, we mainly found GAL co-expressed with POMC and CRH, could be a candidate marker to detect the rare ectopic ACTH^+^&CRH^+^ secreting pheochromocytes. It might be involved in the regulation of the hypothalamic-pituitary-adrenal axis. (Page 7 line 175-182, Figure 3, Figure 4). Second, applied the SCENIC pipeline, we found an additional weak signal of transcription regulons for the DP cells (Page 6 line 153-157, Appendix 1—figure 4). It showed *XPBP1* as the specific regulons for ACTH^+^&CRH^+^ pheochromocyte and adrenocortical cell type. Third, the spliced vs. unspliced phase for CHGA, CHGB, and TH from RNA velocity analysis demonstrated a clear more dynamics expression in POMC+&CRH^+^ pheochromocytes than in pheochromocytes (Appendix 1—figure 5). Lastly, to investigate the genetic driver, the whole exome sequencing identified 1 shared somatic variant of *ACAN* (*c.5951T>A:p.L1984Q*) comparing variants in tumor samples to controls but Sanger sequencing only confirmed the presence in esPHEO_T3 which not observed in esPHEO_T2 (Page 13 line 352-358, Appendix 1—figure 7).

5. The pseudo-time analysis indicated that sustentacular cells transform to ACTH^+^&CRH^+^ pehochromocytes and then to pheochromocyte. The authors utilized Monocle3 in which user has to define the starting points. The authors can validate the result using RNA velocity analysis which also predicts cell transition without the need of prior knowledge about starting point cell type.

At first, we have added RNA velocity analysis (Figure 5B, Page 10 line 268-286). For the high proportions of unspliced/spliced transcripts in Figure 5B, stem-like characteristics of sustentacular cells were supported. We performed the pseudo-time analysis for the adrenal tumor cell subsets to determine the pattern of the dynamic cell transitional states. Then, we re-run the pseudo-time analysis and used the recommended strategy of Monocel to order cells based on genes that differ between clusters. The sustentacular cells were also in an early stage (Figure 6).

6. Given the diverse immune and endothelial cell type in the tumor microenvironment, it would be interesting to perform the cell-cell interaction analysis using the programs such as CellPhoneDB to see if they have distinct regulatory role in different tumor microenvironment.

To investigate the potential cell-cell interactions among various immune cells, endothelial cells, and other cell types in the different tumor microenvironment (esPHEO, esPHEO_Adj, PHEO, and ACA), we performed additional analysis using the CellPhoneDB Python package in the revised version of our manuscript. As shown in the new Appendix 1—figure 6, we observed very distinct patterns of ligand-receptor pairs for cell-cell interactions in the different tumor microenvironments. Notably, the diverse cell clusters within PHEO tumors exhibited a relatively high abundance of cell-cell connections between different cell types, while the cell-cell interactions within esPHEO_Adj samples were totally different. For example, MIF, one of the most enigmatic regulators of innate and adaptive immune responses, was shown as a specific regulator in esPHEO and PHEO, in contrast to ACA.

7. How did the authors define the four subclusters of endothelial cells? Please consider providing list of marker genes.

The four groups of endothelial cells were combined to a larger endothelial cell cluster for downstream analysis. Endothelial cell cluster statistics were added in Supplementary File 3, which presented cell counts for each endothelial cell cluster in different samples and top10 gene markers.

8. In the method part, how did the authors determine different criteria for the maximum number of genes (no more than 5000, 3000, and 2500 genes for PHEO, ACA, and esPHEO samples, respectively)?

We set the different criteria for the maximum number of genes (no more than 5000, 3000, and 2500 genes for PHEO, ACA and esPHEO samples respectively) based on QC violin plot showing the number of detected genes (Appendix 1—figure 2B).

Reviewer #3:Zhang et al. perform single cell RNA sequencing (scRNA-Seq) of one rare ACTH^+^CRH-secreting phenochromocytoma (3 anatomically distinct sites from the tumor and one peritumoral site), one typical pheochromocytoma, and two typical adrenocortical adenomas.Their main findings are as follows: (1) They identify a unique cell type, which they term ACTH^+^CRH^+^ pheochromocyte, which appears to be the tumor cell present in the rare ACTH^+^CRH^+^ tumor (2) Marker gene analysis reveals that while known adrenal chromaffin markers (CHGA, PNMT) are present in both pheochromocytes and ACTH^+^CRH^+^ pheochromocyte, the latter has some unique markers such as GAL and POMC. They validate the marker genes with IHC. (3) Profiling of the non-tumor populations reveals distinct immune microenvironment profile and endothelial cell profile to the rare tumor compared with classical pheochromocytoma and adrenalocortical adenoma.The main strength of this manuscript is that it involves single-cell profiling of an exceptionally rare tumor type and a distinction from the more common adrenal tumors (pheochromocytoma and adrenocortical adenoma). The broader implication of the authors' findings is with respect to Dale's principle, which states that a given neuron releases only one type of neurotransmitter. However, in the case of this tumor, single cell analysis clearly shows that ACTH, CRH, and chatacholemines are being released from the same cell. This is quite interesting and significant. The data will also potentially be valuable to others in the field for analysis in future studies.There remain some unanswered questions – namely:(1) What is the cell in normal physiology that gives rise to this ACTH^+^CRH^+^ pheochromocytoma?(2) Do conventional phenochromocytomas differ from the ACTH^+^CRH^+^ pheochromocytoma in terms of the cell of origin that is transformed, or in the spectrum of genetic alterations that result in transformation?

We thank the reviewer for carefully reviewing the manuscript and raising insightful questions. To response the reviewer’s questions and strengthen the manuscript, we supplemented analysis and experiments as much as possible.

First, we performed RNA velocity analysis (Figure 5, Page 10 line 268-286) to investigate dynamic information in individual cells. For the high proportions of unspliced/spliced transcripts in Figure 5B, stem-like characteristics of sustentacular cells were supported. Also, the spliced vs. unspliced phase for CHGA, CHGB, and TH from RNA velocity analysis demonstrated a clear more dynamics expression in POMC+&CRH^+^ pheochromocytes than in pheochromocytes (Appendix 1—figure 5).

Second, we re-run the pseudo-time analysis (Page 10 line 288-300) and used the recommended strategy of Monocel to order cells based on genes that differ between clusters. The sustentacular cells were also in an early state (Figure 6), which was in accordance with their exhibited stem-like properties and the highest unspliced proportion among non-immune cell types in the RNA velocity analysis (Figure 5B). The results also showed a transition from sustentacular cells to pheochromocytes and then to ACTH^+^&CRH^+^ pheochromocyte, and adrenocortical cells were on another branch (Figure 6). As we discussed in manuscript (Page 14 line 391-398), although pheochromocyte was prior to ACTH&CRH secreting pheochromocyte in pseudotime order, we assumed that ACTH&CRH secreting pheochromocyte have more hormone-producing functions, retain stem- and endocrine-diﬀerentiation ability. But further experiments are needed to validate our hypothesis.

Third, we applied SCENIC analysis pipeline (Page 6 line 153-157, Appendix 1—figure 4) to detect the transcription factors (which are jointly called regulons) alongside their candidate target genes, and yield specific regulons for each cellular cluster. We observed an additional weak signal of transcription regulons (*XPBP1*) for the ACTH^+^CRH^+^ pheochromocytoma and adrenocortical cell type.

Furthermore, to investigate the genetic driver, we supplemented with the whole-exome sequencing (WES) experiments for all rest of tissue samples (esPHEO_T2, esPHEO_T3 and esPHEO_Adj) from the rare case with ectopic ACTH&CRH-secreting pheochromocytoma and the blood sample (esPHEO_Blood). Based on WES data, we identified 1 shared somatic variant of *ACAN* (*c.5951T>A:p.L1984Q*) comparing variants in tumor samples to controls but Sanger sequencing only confirmed the presence in esPHEO_T3 which not observed in esPHEO_T2 (Page 13 line 352-358, Appendix 1—figure 7).

Overall, additional analyses and experiments have presented more comprehensive results which appropriately address the questions raised by the reviewer. But they also provide new hypothesis remaining unanswered questions. For future studies, on one hand, we are very concerned about similar suspicious cases in the clinic. On the other hand, we are going for following research for further downstream experiments to validate the molecular mechanism for secreting multiple hormones.

Comments for the authors:Overall, I think this study is of broad interest given the rarity of this tumor type. My comments to the authors to improve the manuscript are as follows:1. Given how rare the ACTH^+^CRH^+^ pheochromocytoma is, I think the study would be substantially strengthened if the authors could perform DNA sequencing (WGS or WES) and describe how, if at all, the genomic landscape differs from conventional pheochromocytoma.

The frozen tissue of esPHEO_T1 and PHEO_T is unavailable and a few remaining for esPHEO_T2. For all rest of tissue samples, we supplemented with the whole-exome sequencing experiments for tumors (esPHEO_T2, esPHEO_T3) and controls (esPHEO_Adj) from the rare case with ectopic ACTH&CRH-secreting pheochromocytoma. (Page 13 line 352-358, Appendix 1—figure 7)

2. Can the authors comment on whether the hypothesis is whether the ACTH^+^CRH^+^ pheochromocytoma originates from a rare progenitor cell that is distinct from the chromaffin cell giving rise to pheochromocytoma? If so, can the authors stain a panel of normal adrenal glands with some of their marker genes to try and identify this cell in normal tissues?

(Page 14 line 389-398) The RNA velocity estimation and pseudo-time analysis of different adrenal cell subtypes supported the sustentacular cells exhibiting stem-like properties. Although pheochromocyte was prior to ACTH&CRH secreting pheochromocyte in pseudotime order, the RNA velocity prediction of POMC+&CRH^+^ pheochromocytes might be under-estimated because the transcripts of POMC and CRH were all predicted as spliced ones. Based on the spliced vs. unspliced phase for CHGA, CHGB and TH it showed a clear more dynamics expression in POMC+&CRH^+^ pheochromocytes than in pheochromocytes. We assumed that ACTH&CRH secreting pheochromocyte have more hormone-producing functions, retain stem- and endocrine-diﬀerentiation ability. But further experiments are needed to validate our hypothesis.

We thank the reviewer for raising good recommendations. We would like to test marker genes in normal tissues. But it is difficult to obtain normal adrenal glands in clinic. We searched POMC, CRH and GAL in Genotype-Tissue Expression Project (GTEx), which launched by the National Institutes of Health (NIH). GTEx has established a database (https://www.gtexportal.org/home/) to study genes in different normal tissues. The results, as shown in Author response images 1-3: POMC is over-expressed in pituitary, but expressed at a very low level in adrenal gland. CRH is overexpressed in brain-hypothalamus, but almost not expressed in adrenal gland. GAL is overexpressed in pituitary and brain-hypothalamus, but almost not expressed in adrenal gland.

**Author response image 1. sa2fig1:** 

**Author response image 3. sa2fig3:** 

3. While the tumor type is interesting for its rarity, the analysis performed is quite standard and comes across as a bit superficial in parts. Although it is understandable that the authors have only one ACTH^+^CRH^+^ sample I think they can do more with the data and this would significantly strengthen the manuscript. For example, it would be interesting if the authors can point to specific master regulatory factors that drive the distinct programs in pheochromocytes vs. ACTH^+^CRH^+^ pheochromocytes. The immune microenvironment analysis, while inherently descriptive, is also somewhat superficial.

Based on the routine differentially expressed genes analysis, we mainly found GAL co-expressed with POMC and CRH, could be a candidate marker to detect the rare ectopic ACTH^+^&CRH^+^ secreting pheochromocytes. As previous research reported, it might be involved in the regulation of the hypothalamic-pituitary-adrenal axis. (Page 7 line 175-182, Figure 3, Figure 4). Second, applied the SCENIC pipeline, we found an additional weak signal of transcription regulons for the DP cells (Page 6 line 153-157, Appendix 1—figure 4). It showed *XPBP1* as the specific regulons for ACTH^+^&CRH^+^ pheochromocyte and adrenocortical cell type. Furthermore, RNA velocity analysis (Appendix 1—figure 5) demonstrated a clear more dynamics expression in POMC+&CRH^+^ pheochromocytes than in pheochromocytes.

[Editors' note: further revisions were suggested prior to acceptance, as described below.]

Reviewer #2:Although the authors have satisfactorily addressed most of my points, there are remaining concerns about RNA velocity data.Please cite any reference for the statement "For the high proportions of unspliced/spliced transcripts, stem-like characteristics of sustentacular cells were supported." Can global ratio of unspliced/spliced transcripts support stem-like characteristics?Please elaborate Figure 5 C-F. Currently, they don't seem to add any information.

(Page 10 line 269-286, Figure 5 and its legend) We thank the reviewer for carefully reviewing and raising this concern about RNA velocity. We have revised our manuscript to add a paragraph and cite the appropriate references in the updated revision. Previously study had observed that the unspliced transcripts were enriched in genes involved in DNA binding and RNA processing in hematopoietic stem cells [1]. And Schwann cell precursors, which can differentiate into chromaffin cells, also had positive unspliced-spliced phase portrait [2]. Therefore, we claimed that, as for the high proportions of unspliced/spliced transcripts, stem-like characteristics of sustentacular cells were supported.

We remove Figure 5 C-D, as the reviewer mentioned, because they don't seem to add any valuable information. Besides, we added more description about the results for new Figure 5 C-D (old Figure 5 E-F) in Page 10 line 282-288, which showed estimated pseudo-time grounded on transcriptional dynamics and velocity streamlines accounting for speed and direction of motion. These results indicated that medullary cells are earlier than cortical cells (new Figure 5C). From velocity streamlines (new Figure 5D), we found the four adrenal cell subtypes, that is, POMC+&CRH^+^ pheochromocytes, pheochromocytes adrenocortical cells, and sustentacular cells, were independent respectively and not directed toward other cell types.

Reviewer #3:In the revised manuscript Zhang et al. have included additional data and analyses including more exhaustive QC, RNA velocity analysis, regulome analysis, and have performed WES of the ACTH/CRH-secreting pheochromocytoma. They have generally addressed my technical concerns from the prior review. I maintain that the analysis remains somewhat superficial and descriptive in parts and this may be somewhat of a missed opportunity to more deeply explore the underlying biology of this unique case, understanding the caveats of its rarity. Nonetheless, I think a description of this tumor at single-cell resolution and availability of the dataset is of value to the scientific community.However, I would like to see a more careful analysis of the WES data prior to publication. I do not see any basic metrics (mutation rate etc.), description of pathogenicity filtering/annotation, or copy number analysis. The mutations shown are primarily missense and I do not really see any obvious driver genes – how many of these are putative driver vs. passenger mutations? ACAN is mentioned, but what is its significance, if any? The somatic landscape should be discussed in comparison to typical phenochromocytomas and adrenocortical carcinomas, which have been more extensively sequenced. If there is no obvious genetic driver of this ACTH/CRH-secreting phenochromocytoma, that should be stated. If the claim is that ACAN alterations are somehow related to this tumor type, that needs to be substantiated. Or if the implication is that ACAN is a passenger alteration, that needs to be stated explicitly also.

(Page 13 line 359-378; Page 21 line 587-597; Supplementary File 4) We thank the reviewer for carefully reviewing and raising concerns about our WES analysis.

We supplemented the variants filtering criteria in Page 21 line 587-597, and further discussed the WES results in Page 13 line 359-378. Besides, the germline and somatic mutations were listed in Supplementary File 4 including detailed annotations.

Genetic mutations of phaeochromocytoma and paraganglioma are mainly classified into two major clusters, that is, pseudo hypoxic pathway and kinase signaling pathways [3-4]. We did not find any gene mutations or copy number variations that were related to these two major clusters. We only identified 1 shared somatic variant of *ACAN* mutation (c.5951T>A:p.L1984Q) comparing variants in tumor samples to controls. *ACAN*, encoding a major component of the extracellular matrix, is a member of the aggrecan/versican proteoglycan family. Mutations of *ACAN* were reported related to steroid levels [5]. It is well-established that circulating steroid levels are linked to inflammatory diseases such as arthritis, because arthritis as well as most autoimmune disorders result from a combination of several predisposing factors including the stress response system such as the hypothalamic-pituitary-adrenocortical axis [6]. But no direct evidence related to *ACAN* for phaeochromocytoma. Therefore, no obvious genetic driver was found to explain the rare case of ACTH/CRH-secreting phaeochromocytoma. Further investigations would be needed to uncover the relation between *ACAN* to phaeochromocytoma.

References:

[1]. Bowman TV, McCooey AJ, Merchant AA, Ramos CA, Fonseca P, Poindexter A, Bradfute SB, Oliveira DM, Green R, Zheng Y, Jackson KA, Chambers SM, McKinney-Freeman SL, Norwood KG, Darlington G, Gunaratne PH, Steffen D, Goodell MA. Differential mRNA processing in hematopoietic stem cells. Stem Cells. 2006. Mar;24(3):662-70.

[2]. La Manno G., Soldatov R., Zeisel A., Braun E., Hochgerner H., Petukhov V., Lidschreiber K., Kastriti M.E., Lönnerberg P., Furlan A. RNA velocity of single cells. Nature. 2018 560:494-498.

[3] Pillai S, Gopalan V, Smith RA, Lam AK. Updates on the genetics and the clinical impacts on phaeochromocytoma and paraganglioma in the new era. Crit Rev Oncol Hematol. 2016. Apr;100:190-208.

[4] Nölting S, Grossman AB. Signaling pathways in pheochromocytomas and paragangliomas: prospects for future therapies. Endocr Pathol. 2012. Mar;23(1):21-33.

[5] Yousri NA, Fakhro KA, Robay A, Rodriguez-Flores JL, Mohney RP, Zeriri H, Odeh T, Kader SA, Aldous EK, Thareja G, Kumar M, Al-Shakaki A, Chidiac OM, Mohamoud YA, Mezey JG, Malek JA, Crystal RG, Suhre K. Whole-exome sequencing identifies common and rare variant metabolic QTLs in a Middle Eastern population. Nat Commun. 2018 Jan 23;9(1):333.

[6]. Cutolo M, Sulli A, Pizzorni C, Craviotto C, Straub RH. Hypothalamic-pituitary-adrenocortical and gonadal functions in rheumatoid arthritis. Ann N Y Acad Sci. 2003 May;992:107-17.